# Measurement report: Intra- and interannual variability and source apportionment of volatile organic compounds during 2018–2020 in Zhengzhou, central China

Shijie Yu[TS1][1,2], Shenbo Wang[2,3], Ruixin Xu[2,3], Dong Zhang[1,2], Meng Zhang[5], Fangcheng Su[2,3], Xuan Lu[1,2], Xiao Li[2,3], Ruiqin Zhang[2,3], and Lingling Wang[4]

[1]College of Chemistry, Zhengzhou University, Zhengzhou 450001, China
[2]Institute of Environmental Sciences, Zhengzhou University, Zhengzhou 450001, China
[3]School of Ecology and Environment, Zhengzhou University, Zhengzhou 450001, China
[4]Environmental Monitoring Center of Henan Province, Zhengzhou 450000, China
[5]Pingdingshan Ecological Environment Monitoring Center of Henan Province, Pingdingshan 467000, China

**Correspondence:** Ruiqin Zhang (rqzhang@zzu.edu.cn) and Lingling Wang (lindaw2007@126.com)

**Abstract.** Ambient volatile organic compounds (VOCs) were measured continuously from January 2018 to December 2020 at an urban site in Zhengzhou (China) to investigate their characteristics, sources, atmospheric oxidation capacity (AOC), and chemical reactivity. During the sampling period, the total concentration of observed VOCs was $94.3 \pm 53.1 \, \mu g \, m^{-3}$, and alkanes were the major VOC species, accounting for 58 % of the total. During the sampling period, the interannual variation in VOCs gradually reduced from $113.2 \pm 65.2 \, \mu g \, m^{-3}$ in 2018 to $90.7 \pm 52.5 \, \mu g \, m^{-3}$ in 2019 and $79.1 \pm 41.7 \, \mu g \, m^{-3}$ in 2020. Ethane and propane were the top two most abundant species during the 3-year observation period. Results showed that the total AOC, dominated by OH radical reactions, was $7.4 \times 10^7$ molec. $cm^{-3} \, s^{-1}$. Total OH reactivity was $45.3 \, s^{-1}$, and it was mainly contributed by $NO_x$. The AOC and ·OH reactivity both exhibited well-defined seasonal and interannual patterns. Therefore, control strategies should focus on the key species given their interannual and seasonal variations. Meanwhile, diagnostic ratios of VOC species indicated that VOCs in Zhengzhou were greatly affected by vehicle emissions and liquid petroleum gas/natural gas (LPG/NG). Positive matrix factorization analysis identified six sources: industrial sources, solvent use, vehicle emissions, LPG/NG, fuel burning, and biogenic sources. Vehicle emissions and industrial sources made the largest contributions to VOC emissions in each of the 3 years. The proportion of the contributions of vehicle emissions and LPG/NG increased with each passing year. However, the proportion of industrial and solvent sources presented a decreasing trend, which reflects the remarkable effect of control policies. The effect of VOCs on $O_3$ formation suggests that vehicle emissions and solvent use remain key sources. Therefore, it is necessary to formulate effective strategies for reducing ground-level $O_3$, and those sources mentioned above should be strictly controlled by the regulatory authorities.

## 1  Introduction

In recent years, regional atmospheric pollution events have occurred frequently around the world, with many areas suffering severe haze episodes in autumn and winter, as well as $O_3$ pollution in summer (Uttamang et al., 2020; B. Li et al., 2019; Sadeghi et al., 2022; Yan et al., 2018; Yadav et al., 2019; Zhang et al., 2018). Volatile organic compounds (VOCs) are important precursors of secondary pollutants such as $O_3$ and secondary organic aerosols, and the study of VOCs is a primary focus among the scientific community and relevant governing bodies (Liu et al., 2019a; S. Song et al., 2019; Xu et al., 2017).

VOCs encompass a large variety of species, and the chemical reactivity of each species varies greatly. Thus, elucidation of VOC characteristics and active substance identification represent research priorities. In many regions, alkanes represent the dominant VOC species, while studies which do not report volatile organic VOCs (OVOCs) usually identify aromatics and alkenes as better contributors of ozone formation potential (OFP) (K. Li et al., 2019; Yan et al., 2017). Following a study in Wuhan (China), Yang et al. (2019) suggested that alkanes were the dominant group, accounting for 42 % of the total VOC concentration, and that $C_2$–$C_3$ hydrocarbons were the dominant active substance. Huang and Hsieh (2019) studied the maximum incremental reactivity (MIR) and propylene-equivalent (PE) concentration, reporting that toluene was the largest potential contributor to the OFP and that industrial emissions contributed nearly 60 % to the OFP. Given the complex composition of VOCs in the atmosphere, it is difficult to determine the sources of VOCs. To apportion the source contributions of VOCs, it is common practice to use receptor models that include positive matrix factorization (PMF), chemical mass balance, and principal component analysis. In China, traffic emissions are often the main source of VOCs, particularly in major metropolitan areas (B. Li et al., 2019; Liu et al., 2019b; M. Song et al., 2019). Moreover, industrial processes and solvent use have remarkable influence on VOC emissions (Hui et al., 2019; Mo et al., 2017). Biogenic sources also cannot be ignored owing to their high reactivity, which can contribute 5 %–20 % of VOC emissions (Wu et al., 2016). In addition to the study of VOC characteristics and source apportionment, analysis of atmospheric oxidation characteristics is another hot topic area. Variations in the atmospheric oxidation capacity (AOC) not only affect the $O_3$ level in summer but also greatly impact the generation of secondary particles throughout the entire year (Prinn, 2003). However, most previous related studies in China focused on metropolitan areas such as the Beijing–Tianjin–Hebei region (Y. Gu et al., 2019), Pearl River Delta region (Zhang et al., 2015), and Yangtze River Delta region (Xu et al., 2017; Zheng et al., 2020), while less research has been conducted in heavily polluted areas of the central plains. Moreover, previous studies frequently used short-term monitoring data that cannot fully reflect the comprehensive VOC pollution char-

acteristics within a region. Therefore, it is an urgent requirement that large-scale investigations be conducted on the central plains of China.

As the political and cultural center of Henan Province, Zhengzhou had more than 10 million permanent residents and 4.0 million private vehicles in 2019 (X. Gu et al., 2019). This area is often confronted with severe haze episodes and photochemical pollution due to the emission of huge amounts of air pollutants (Y. Li et al., 2020). Consequently, much research has been conducted on air pollution in Zhengzhou, but many of the earlier studies focused only on particulate matter (Jiang et al., 2018; Wang et al., 2019). Few studies have addressed VOCs in Zhengzhou, especially in relation to longer sampling periods and seasonal measurements. Following adjustment of control policies and optimization of the energy structure, recent mitigation measures might have affected the characteristics and sources of VOCs. Therefore, it is necessary to identify local VOC pollution characteristics on the basis of a long time series of monitoring data, which could provide a reference for the formulation of local air pollution control measures.

To deepen the understanding of VOC pollution characteristics, chemical reactivity, and source contribution, VOC data were measured in Zhengzhou using online instruments from 2018 to 2020. The aims of this research were as follows: (1) analyze the characteristics of VOC variation in Zhengzhou, including diurnal, seasonal, and annual changes; (2) quantify the contribution of sources among intra- and interannual variations and identify the locations of VOC sources; (3) parameterize the AOC and speciate OH reactivity; and (4) assess $O_3$ formation by MIR and PE analysis.

## 2  Methodology

### 2.1  Study site and measurements

The observation period of this study was from January 2018 to December 2020. The VOC samples were collected at an urban site (34°45′ N, 113°41′ E) by the Department of Environmental Protection of Henan Province. The surrounding environment represents a typical urban environment in Zhengzhou, comprising commercial and residential districts with traffic sources (Fig. 1).

The VOC species were monitored continuously using an autoGC system with a 1 h time resolution (AMA Instruments GmbH, Germany). Specific information regarding this system is described in Zou et al. (2015). The 57 VOC species (comprising 29 alkanes, 10 alkenes, 1 alkyne, and 17 aromatics) were calibrated using the VOC standards of the U.S. EPA Photochemical Assessment Monitoring Stations (PAMS) mixture (Spectra Gases, USA) before monitoring. During the observation period, zero and span gas checks (PAMS calibration gases) were conducted monthly using the five-point method, together with the adjustment of the retention time. Data quality control of this instrument is detailed

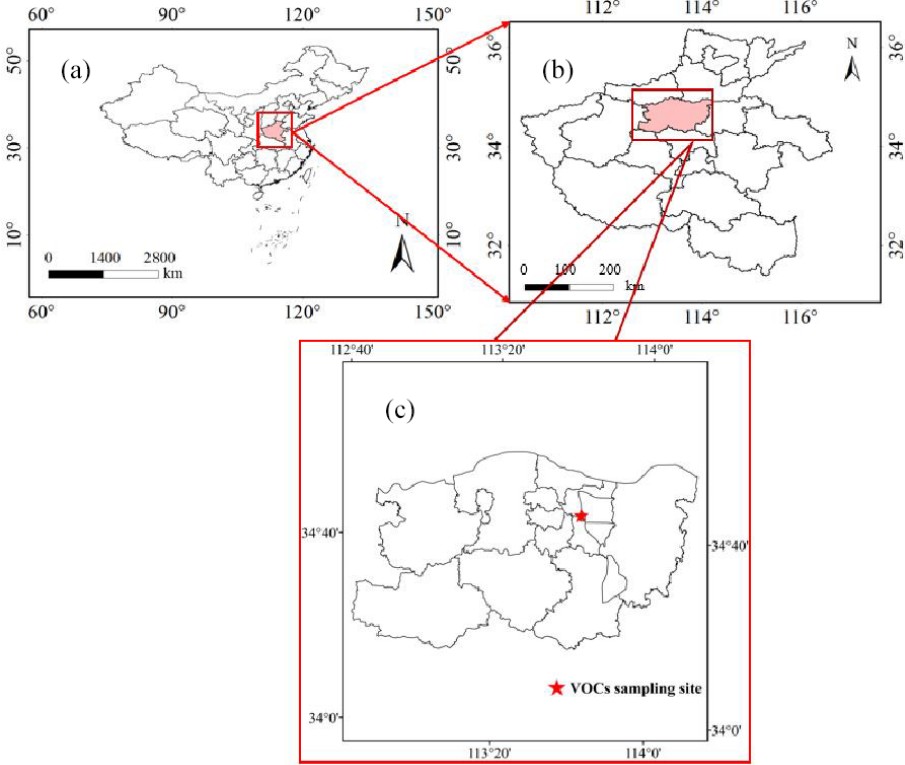

**Figure 1.** Locations of the sampling stations. **(a)** China, **(b)** Henan, and **(c)** Zhengzhou.

in our previous study (Ren et al., 2020). Overall, the correlation coefficient varied from 0.990 to 0.999, and the detection limits were in the range of 0.02–0.12 ppbv, as shown in Table S1 in the Supplement.

Meteorological data comprising atmospheric temperature ($T$), relative humidity, ultraviolet (UV) radiation, precipitation (Pr), planetary boundary layer height, wind speed (WS), and wind direction (WD) were obtained from the surface meteorological station of the Henan monitoring center. Datasets of trace gases such as the hourly concentrations of NO, $NO_2$, CO, $O_3$, $SO_2$, and fine particles ($PM_{2.5}$) were obtained from the ambient air quality observation station of Yanchang, which is located approximately 2 km from the VOC monitoring station.

## 2.2   Positive matrix factorization (PMF) model

In this study, analysis of the source of the VOCs was performed using the EPA PMF 5.0 model, which is a receptor model used widely for source apportionment (Gao et al., 2018; Yadav et al., 2019). Detailed information regarding this method is available in the user manual (Norris et al., 2014) and other related literature (C. Song et al., 2019; M. Song et al., 2019). Two input files are required for PMF: the concentration values and the uncertainty values of the individual VOC species. The uncertainty is calculated using Eq. (1) when the species concentration value is higher than

its method detection limit (MDL) or using Eq. (2) when the concentration is less than or equal to the MDL:

$$Unc = \sqrt{(EF \times c)^2 + (0.5 \times MDL)^2}, \tag{1}$$

$$Unc = 5/6 \times MDL, \tag{2}$$

where $c$ is the concentration of the individual VOC species, and EF is the error fraction, which was set to 10 % of the VOC concentration (Yuan et al., 2012).

Owing to the complexity of the chemical reactions, not all of the VOC species were used in the PMF analysis. Based on previous work, this study adopted the following principles for selection of the VOC species. (1) Species with more than 25 % of data missing or below the MDLs were rejected, which follows the methodology of previous studies (Zhou et al., 2019). (2) Species with short atmospheric lifetimes were excluded because they rapidly react away in the atmosphere. (3) Species that represent source tracers of emission sources were retained (e.g., in the case of isoprene). Eventually, 27 VOC species were selected for source apportionment analysis. VOC species were grouped into strong, weak, and bad according to their signal / noise ratio ($S/N$), and there were 23 and 4 species grouped into strong and weak, respectively. It should be noted that the volumetric concentration (ppbv) of the individual VOC species was converted to mass concentration ($\mu g\,m^{-3}$) before being input into the PMF model.

Choosing the optimal number of factors in the model is important. The number of factors depends on $Q$ (ture) $/Q$ (robust) and $Q/Q$ expected ($Q_{exp}$). In theory, $Q$ (ture) $/Q$ (robust) $< 1.5$ and a value close to 1 are considered reasonable (Ulbrich et al., 2009), and the rate of change in $Q/Q_{exp}$ should be stable, and the ratio should be close to 1 (Baudic et al., 2016; Hui et al., 2019). In this study, the numbers of factors used for the PMF analysis were tested from three to eight, and the optimum six-factor solution with $Q/Q_{exp} = 0.94$ ($Q$ (ture) $/Q$ (robust) $= 1.0$) was selected. Additionally, $F_{peak}$ values from $-1$ to 1 with 0.1 intervals were used in the model, and $F_{peak} = -0.2$ was established as the best solution (as shown in Fig. S1).

## 2.3 Relative reactivity of VOCs

To better understand the role of VOCs in the formation of tropospheric $O_3$, the OFP and the PE concentration were investigated to analyze the chemical reactivity of the VOC species (Carter, 1994; Atkinson and Arey, 2003), and the detailed operation is described in the Supplement (Sect. S1).

## 2.4 Atmospheric oxidation capacity (AOC) and speciated oxidant reactivity

The AOC is defined as the sum of the respective oxidation rates of the primary pollutants (e.g., CO, $CH_4$, and VOCs) by the oxidants (e.g., OH, $NO_3$, and $O_3$), and it can be calculated using Eq. (1) (Elshorbany et al., 2009; Xue et al., 2016):

$$AOC = \sum_i k_{Y_i}[X][Y_i],$$ (3)

where $[X]$ and $[Y_i]$ are the number concentrations of molecule oxidant $X$ and $Y_i$, respectively, and $k_{Y_i}$ is the bimolecular rate constant of molecule $Y_i$ with oxidant $X$ (Zhu et al., 2020). In this study, the reduced substances only included 57 PAMS (provided by Spectra Gases Inc., USA) and CO. The oxidants only included ·OH, $NO_3$, and $O_3$ radicals. The concentration OH and $NO_3$ radicals was estimated from parameterization methods via Eqs. (4) and (5) (Carter, 1994; Warneke et al., 2004):

$$[OH] = a \times (J_{O^1D})^\alpha \times (J_{NO_2})^\beta$$
$$\times \frac{b \times [NO_2] + 1}{c \times [NO_2]^2 + d \times [NO_2] + 1},$$ (4)

where $J_{O^1D}$ and $J_{NO_2}$ are the measured photolysis frequency $(s^{-1})$ of $O_3$ and $NO_2$, respectively. The values of $a$, $b$, $c$, $\alpha$, and $\beta$ are $4.1 \times 10^9$, 140, 0.41, 1.7, 0.83, and 0.19, respectively (Yang et al., 2019).

$NO_3$ concentration in the atmosphere is based on the steady-state assumption (Liebmann et al., 2018):

$$[NO_3]$$
$$= \frac{k_{NO_2} \times [NO_2] \times [O_3]}{J_{NO_3} + J_{NO+NO_3} \times [NO] + \sum_i k_{NO_3} + VOC_i \times [VOC]_i},$$ (5)

where $J_{NO_3}$ is the measured photolysis frequency $(s^{-1})$ of $NO_3$. The rate coefficients for $NO_2$–$O_3$ ($k_{NO_2+O_3}$) and $NO$–$NO_3$ ($k_{NO+NO_3}$) were obtained from Atkinson et al. (2004). Detailed descriptions of the calculation processes of OH and $NO_3$ are available in Yang et al. (2021).

Additionally, OH reactivity is another indicator of atmospheric oxidation. It is the inverse of the OH lifetime, and it can be defined as the product of the rate coefficients and the concentrations of the reactants with OH. Thus, OH reactivity can be calculated using Eq. (6) (Mao et al., 2010):

$$OH \text{ reactivity} = \sum k_{OH+VOC_i} \times [VOC_i] + k_{OH+CO}$$
$$\times [CO] + k_{OH+NO}$$
$$\times [NO] + k_{OH+NO_2}$$
$$\times [NO_2] + k_{OH+SO_2}$$
$$\times [SO_2] + k_{OH+O_3}$$
$$\times [O_3] + k_{OH+other} \times [other],$$ (6)

where $[X_i]$ is the concentration of the species (e.g., CO, $NO_x$, $SO_2$, and VOCs), and the rate coefficient $k_{OH}$ (unit: $cm^3 molec.^{-1} s^{-1}$) represents the corresponding reaction rate coefficient.

## 2.5 Conditional probability function (CPF) analysis

The conditional probability function (CPF) was developed to identify potential source contributions using the PMF source contribution solution, coupled with WD (Guo et al., 2011; Hsu et al., 2018; Wu et al., 2016). The CPF is defined as follows:

$$CPF = \frac{m_{\Delta\theta}}{n_{\Delta\theta}},$$ (7)

where $m_{\Delta\theta}$ is the number of appearances from wind sector $\Delta\theta$ (each sector is 22.5°) that exceed the concentration threshold (75th percentile of each source contribution), and $n_{\Delta\theta}$ is the total number of occurrences in the same wind sector. Weak winds (WS $< 1.5 \text{ m s}^{-1}$) were excluded from the calculation because of the difficulty in defining WD (Zheng et al., 2018).

## 3 Results and discussion

### 3.1 Characteristics of VOCs in Zhengzhou

#### 3.1.1 Concentrations and compositions of VOCs

The average ambient VOC concentrations and chemical species measured in Zhengzhou during the study period are shown in Table S2. Table S3 presents the comparison of VOCs between this study and other studies. The annual average concentration of VOCs was $94.3 \pm 53.1 \text{ μg m}^{-3}$ ($38.2 \pm 15.6$ ppbv), i.e., close to the concentration reported

in Langfang (33.4 ppbv) (C. Song et al., 2019) and Nanjing (34.4 ppbv) (Shao et al., 2016), lower than that found in Chengdu (41.8 ppbv) (Song et al., 2018) and Guangzhou (42.7 ppbv) (Zou et al., 2015), and higher than that reported in Tianjin (28.7 ppbv) (Liu et al., 2016). Among the observed species, alkanes were the major component of the VOCs with a mean concentration of $54.7 \pm 37.9\,\mu g\,m^{-3}$, accounting for 58 % of the total, followed by aromatics (25 %), alkenes (13 %), and alkynes (3 %). Many previous related studies also reported that alkanes represent the dominant group (Fu et al., 2020; Gu et al., 2020), similar to the situation found in Zhengzhou. For the record, OVOCs were not simultaneously measured in this study due to the limitations on available instrumentation. Thus, those investigations apply only to studies which failed to measure OVOCs.

To clarify the characteristics of VOC emission sources, the concentrations of the 20 most abundant species, accounting for 83 % of the compound classes monitored in the present study, are listed in Table 1. During the sampling period (2018–2020), the most important VOC species in Zhengzhou were ethane ($11.7 \pm 6.8\,\mu g\,m^{-3}$), propane ($8.2 \pm 4.9\,\mu g\,m^{-3}$), toluene ($6.5 \pm 4.5\,\mu g\,m^{-3}$), $i$-pentane ($6.1 \pm 10.4\,\mu g\,m^{-3}$), $n$-butane ($5.8 \pm 4.1\,\mu g\,m^{-3}$), and ethene ($5.6 \pm 5.3\,\mu g\,m^{-3}$). Generally, $C_2$–$C_5$ species are closely related to vehicular emissions, coal burning, and liquid petroleum gas (LPG) (Hui et al., 2021; Zhang et al., 2020). Among the most abundant 20 VOC species, half were alkanes, accounting for 79 % of the total alkanes measured. The $C_2$–$C_3$ alkanes mainly originate from LPG, while $C_4$–$C_5$ alkanes are considered tracers of vehicle emissions (Fan et al., 2021). The most abundant alkene species were ethylene, propene, isoprene, and propene, representing 72 % of the total alkanes. The $C_2$–$C_3$ alkenes are mainly derived from vehicle emissions and LPG (Zhang et al., 2015), whereas isoprene is a typical biogenic tracer (Maji et al., 2020). Isoprene emissions have also been reported from biomass burning, e.g., from smoldering rice straw fires (Kumar et al., 2021). In the 20 most abundant VOC species, there were six aromatics: toluene, $m/p$-xylene, benzene, ethylbenzene, $o$-xylene, and $p$-diethylbenzene. These aromatics are the most frequently observed aromatic compounds in urban areas, originating from vehicle emissions, industrial processes, solvent use, and combustion sources (Hui et al., 2019).

### 3.1.2 Interannual variation in VOCs

The interannual average concentrations and contributions of VOCs during 2018–2020 are presented in Table S2. The interannual variation in the VOCs declined gradually as follows: $113.2 \pm 65.2\,\mu g\,m^{-3}$ in 2018, $90.7 \pm 52.5\,\mu g\,m^{-3}$ in 2019, and $79.1 \pm 41.7\,\mu g\,m^{-3}$ in 2020. It should be mentioned that the differences in these yearly averages are statistically significant considering the large standard deviations indicated. The trend of decrease in VOCs could be attributed to increasingly stringent policies regarding emission reduction and the influence of the COVID-19 lockdown on air quality in 2020 (M. Wang et al., 2021).

The concentrations of the 20 most abundant species are listed in Table 1. Ethane and propane were the top two most abundant species in each of the 3 years, indicating that LPG and vehicular emissions had substantial impact on the area surrounding the sampling site (Yadav et al., 2019). The $C_4$–$C_5$ alkanes and some aromatics represent the main tracers of motor vehicle emissions (Fan et al., 2021). However, the quantities of those species were diminished in 2020, which might represent a consequence of the COVID-19 pandemic. As a tracer of coal burning, acetylene decreased gradually from $4.8 \pm 4.9\,\mu g\,m^{-3}$ in 2018 to $0.9 \pm 1\,\mu g\,m^{-3}$ in 2020.

### 3.1.3 Seasonal variations

As plotted in Fig. S2 and listed in Table S4, the monthly mean mixing ratios of the VOCs and compounds were investigated. The VOCs showed clear seasonal dependence with the highest concentration in winter ($116.5\,\mu g\,m^{-3}$) followed by spring ($86.4\,\mu g\,m^{-3}$), autumn ($86.1\,\mu g\,m^{-3}$), and summer ($74.2\,\mu g\,m^{-3}$). Meanwhile, the seasonal variation in the group of alkanes, alkenes, alkynes, and aromatics was similar to that of the VOCs. Additionally, the monthly mean mixing ratios of the dominant and tracer species are plotted in Fig. 2. The results show that almost all VOCs had clear seasonal dependence, with the highest concentrations in winter and lowest concentrations in summer. However, the mixing ratio of isoprene was highest ($2.46\,\mu g\,m^{-3}$) in July and lowest ($0.33\,\mu g\,m^{-3}$) in December. As a tracer of biogenic sources, isoprene showed positive correlation with $T$ ($R^2 = 0.61$, $p < 0.01$). In addition to biogenic emissions, the seasonal variation in VOCs was mainly influenced by changes of anthropogenic sources. As a northern city of China, Zhengzhou emits substantial quantities of pollutants during the heating season in winter (Wang et al., 2019). The higher concentrations of VOCs and tracers (such as acetylene and aromatics) in winter might be derived from coal combustion (Zhang et al., 2020).

Seasonal variations in VOC concentrations are associated with several factors such as photochemical activities and meteorological conditions. In summer, VOCs are consumed under the condition of high $T$, strong UV radiation, and high concentration of OH radicals (Huang et al., 2019). In winter, the high level of VOCs can often be attributed to a lower boundary layer and calm weather conditions (Hui et al., 2019). Additionally, the transport of pollutants from the Beijing–Tianjin–Hebei region cannot be ignored because of the north wind that prevails in Zhengzhou in winter. The seasonal variation in VOCs in several cities was investigated, and the variation trend of VOC concentrations in most studies was found to be similar to that observed in Zhengzhou (Liu et al., 2019; Yadav et al., 2019).

**Table 1.** Top 20 most abundant VOC species ($\mu g\,m^{-3}$) measured in Zhengzhou for the study period 2018–2020.

| Species | 2018 | Species | 2019 | Species | 2020 | Species | Average |
|---|---|---|---|---|---|---|---|
| Ethane | $13.4 \pm 7.5$ | Ethane | $11.6 \pm 7.7$ | Ethane | $10.2 \pm 5.3$ | Ethane | $11.7 \pm 6.8$ |
| Propane | $8.6 \pm 5.2$ | Propane | $8.8 \pm 5.6$ | Propane | $7.3 \pm 4$ | Propane | $8.2 \pm 4.9$ |
| *i*-Pentane | $7.9 \pm 19.7$ | *n*-Butane | $6.2 \pm 4.7$ | Toluene | $5.6 \pm 4.3$ | Toluene | $6.5 \pm 4.5$ |
| Toluene | $7.8 \pm 4.3$ | Toluene | $6.1 \pm 4.7$ | Ethene | $5.5 \pm 4.4$ | *i*-Pentane | $6.1 \pm 10.4$ |
| Ethene | $7.6 \pm 6.2$ | *i*-Pentane | $6 \pm 6.8$ | *n*-Butane | $5.4 \pm 3.7$ | *n*-Butane | $5.8 \pm 4.1$ |
| *n*-Pentane | $6 \pm 16.1$ | *m/p*-Xylene | $4.5 \pm 4.4$ | *i*-Pentane | $4.6 \pm 4.7$ | Ethene | $5.6 \pm 5.3$ |
| *n*-Butane | $5.9 \pm 3.8$ | Benzene | $3.9 \pm 2.4$ | *m/p*-Xylene | $4.3 \pm 5.4$ | *m/p*-Xylene | $4.7 \pm 4.3$ |
| *m/p*-Xylene | $5.2 \pm 3$ | Ethene | $3.8 \pm 5.3$ | *i*-Butane | $3.5 \pm 2.2$ | Benzene | $3.8 \pm 2.3$ |
| Acetylene | $4.9 \pm 5$ | *i*-Butane | $3.8 \pm 2.7$ | Benzene | $3.4 \pm 1.9$ | *n*-Pentane | $3.8 \pm 7.3$ |
| Cyclopentane | $4.4 \pm 23.4$ | *n*-Hexane | $3.7 \pm 3.5$ | *n*-Pentane | $2.3 \pm 2.8$ | *i*-Butane | $3.5 \pm 2.4$ |
| Benzene | $4.1 \pm 2.5$ | Acetylene | $3.3 \pm 4.8$ | Ethylbenzene | $2 \pm 2$ | Acetylene | $3 \pm 3.6$ |
| *i*-Butane | $3.1 \pm 2.1$ | *n*-Pentane | $3.1 \pm 2.9$ | Isoprene | $1.9 \pm 3.4$ | Ethylbenzene | $2.4 \pm 1.8$ |
| Ethylbenzene | $3.1 \pm 1.8$ | 3-Methylpentane | $2.6 \pm 1.9$ | *n*-Hexane | $1.1 \pm 1.3$ | *n*-Hexane | $2.2 \pm 2.2$ |
| Isoprene | $2.8 \pm 3.2$ | Ethylbenzene | $2.1 \pm 1.6$ | Styrene | $1.1 \pm 1.7$ | Cyclopentane | $2 \pm 9$ |
| *n*-Hexane | $1.9 \pm 1.6$ | Propene | $1.6 \pm 1.9$ | *o*-Xylene | $1.1 \pm 1.4$ | 1-Isoprene | $1.9 \pm 2.8$ |
| *o*-Xylene | $1.7 \pm 1$ | *o*-Xylene | $1.2 \pm 1.1$ | Propene | $1 \pm 2.2$ | 3-Methylpentane | $1.7 \pm 1.5$ |
| 3-Methylpentane | $1.7 \pm 1.6$ | 2-Methylpentane | $1.1 \pm 1.4$ | *n*-Heptane | $1 \pm 1.6$ | *o*-Xylene | $1.4 \pm 1.2$ |
| Styrene | $1.6 \pm 1.2$ | Isoprene | $1.1 \pm 1.9$ | *m*-Ethyltoluene | $1 \pm 1.2$ | Propene | $1.3 \pm 1.7$ |
| *p*-Diethylbenzene | $1.6 \pm 1.2$ | Cyclopentane | $1.1 \pm 2.2$ | *n*-Undecane | $1 \pm 1.7$ | Styrene | $1.2 \pm 1.3$ |
| Propene | $1.4 \pm 0.9$ | *p*-Diethylbenzene | $1 \pm 0.9$ | *n*-Dodecane | $1 \pm 2.4$ | *p*-Diethylbenzene | $1.2 \pm 1$ |
| $\sum$TOP 20 species/$\sum$VOCs | 84 % | | 84 % | | 81 % | | 83 % |

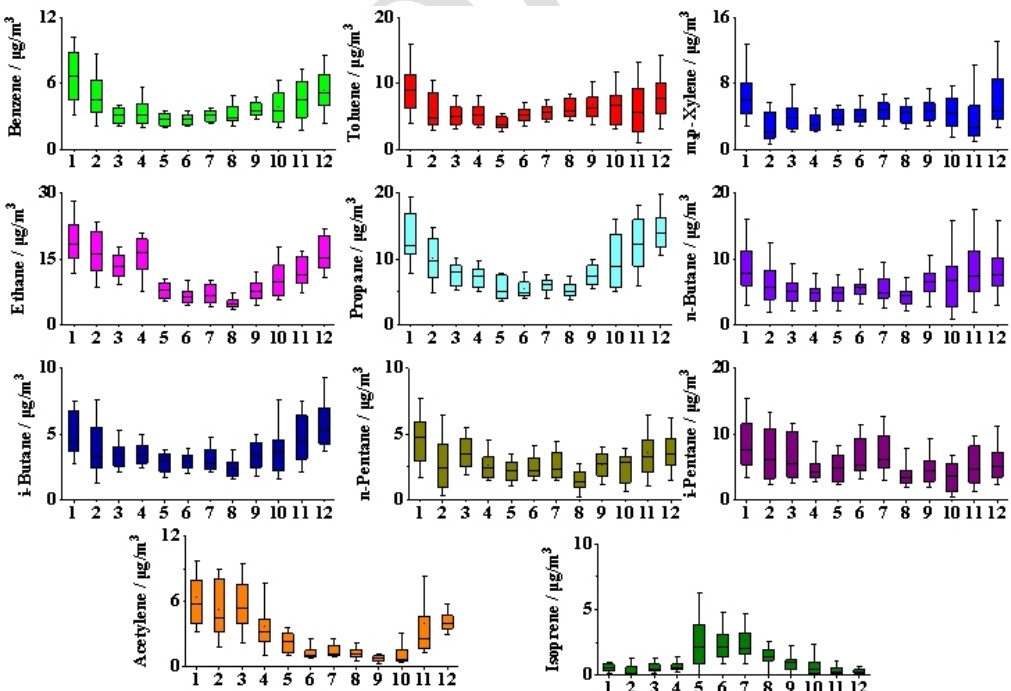

**Figure 2.** Monthly changes in the concentrations of the typical VOC species in Zhengzhou. The upper and lower boundaries of the boxes indicate the 75th and 25th percentiles, respectively; the lines within the boxes mark the median; the whiskers above and below the boxes indicate the 90th and 10th percentiles, respectively.

### 3.1.4 Diurnal variations

The diurnal variations in VOCs, trace gases ($NO_2$ and $O_3$), and meteorological parameters ($T$, relative humidity, WS, and UV) are shown in Fig. S3. The VOCs present negative correlation with $O_3$ ($R^2 = -0.82$, $p < 0.01$), whereas the diurnal variation in VOCs shows moderate consistency with the variation in $NO_2$ ($R^2 = 0.62$, $p < 0.01$). The high values of VOCs generally appeared in the morning with low $O_3$ concentrations. The peak in the morning was attributed to vehicle emission (Li et al., 2019b). Additionally, local meteorological and atmospheric processes also play important roles in the diurnal variation in VOCs in ambient air. In the early morning, the concentrations of VOCs remained high owing to stable atmospheric conditions and a shallow boundary layer height (Hui et al., 2020). The concentration of VOCs declined to the lowest value at 15:00 China standard time (CST), when $O_3$ reached a maximum level with the production and consumption rates of $O_3$ in equilibrium. In the afternoon, higher $T$ and increased UV radiation intensity led to consumption of VOCs. Moreover, higher WS also accelerated the diffusion of VOCs. Subsequently, VOC concentrations gradually accumulated with the arrival of the late traffic peak and remained at a high level throughout the night. It should be noted that VOC concentrations at night were generally higher than those during the day. Previous studies have suggested that VOCs can be oxidized by O, OH radicals, and $NO_3$ radicals (Atkinson and Arey, 2003). During the daytime, reactions with OH radicals and $O_3$ are the most important chemical reactions for VOCs, whereas the reactions with $NO_3$ radicals and $O_3$ are the main sedimentation reactions at night. Concentrations of VOCs were generally higher at night because the chemical activity of OH radicals is much higher than that of $NO_3$ radicals (Carter, 2010).

The mean diurnal variations in high-concentration and tracer VOCs were investigated, as shown in Fig. 3. As tracers of motor vehicle emissions (Zheng et al., 2018), $i$-pentane and $n$-pentane showed remarkable peaks at 08:00 CST that are considered to reflect vehicle emissions. Furthermore, benzene, toluene, ethylbenzene, xylene, and the $C_2$–$C_4$ alkanes also presented similar diurnal variation characteristics that are considered to represent the substantial effect of motor vehicles. However, the mixing ratio of isoprene showed higher values in the afternoon and had a trend similar to that of $T$. The elevated values of isoprene in the afternoon indicated substantial emissions from biogenic sources.

### 3.2 Diagnostic ratios

Some VOC species are commonly used as indicators of emission sources. To characterize the seasonal differences in the contributions of the various sources, this study adopted the benzene / toluene (B / T) ratio and the $i$-pentane / $n$-pentane ratio as the preferred metrics.

The B / T ratio can be used to distinguish potential sources such as traffic emissions, coal plus biomass combustion, and solvent use. The diagnostic ratios varied according to the emission sources (i.e., below 0.20 for solvent use, 0.5 for traffic sources, 1.5–2.2 for coal combustion, and 2.5 for biomass burning) (Huang et al., 2019; Li et al., 2019a). As shown in Fig. S4, the highest value of the B / T ratio was 0.55 in winter, suggesting that traffic emissions affected the ambient atmosphere. The ratios of 0.31, 0.27, and 0.31 in spring, summer, and autumn, respectively, indicate that aromatics were more likely derived from the mixed sources of solvent use and vehicle emissions during these seasons. As a transportation hub, Zhengzhou has a large number of motor vehicles (X. Gu et al., 2019). Therefore, regional controls on motor vehicle use should be strengthened.

The $i$-pentane / $n$-pentane ratio was also investigated. Generally, $i$-pentane and $n$-pentane have similar reaction rates with OH radicals, and the ratio of this pair of species is indicative of different sources. The $i$-pentane / $n$-pentane ratio associated with coal combustion, vehicle emissions, and fuel evaporation is generally 0.56–0.80, $\sim 2.2$, and 3.8, respectively (Huang et al., 2019; Li et al., 2019a). The highest $i$-pentane / $n$-pentane ratio was found in summer (2.79), indicating strong impact from traffic sources. The ratio was 1.88 in spring and 1.96 in autumn, suggesting that most VOCs originated from the mixed sources of vehicle emissions and coal combustion. Notably, the average ratio in winter (1.55) was lower than that in the other three seasons, indicating the stronger contribution from coal burning in the heating season.

### 3.3 Source identification

In this study, six sources were identified using the PMF model: industrial sources, solvent use, vehicle emissions, LPG/natural gas (NG), fuel burning, and biogenic sources. The source profiles of the VOCs during the sampling period are presented in Fig. 4.

### 3.3.1 Factor 1: industrial sources

Factor 1 was characterized by high loadings on ethylene (60.6 %), ethane (31.6 %), propylene (25.8 %), and propane (24.0 %). Ethene and propylene are the first and second most abundant industrially produced organic compounds worldwide, respectively (Eckert et al., 2014). Additionally, this factor was also dominated by aromatics such as styrene (76.0 %), toluene (34.3 %), and ethylbenzene (28.1 %), which are typical tracers of industrial sources. Meanwhile, this source had correlation with gas tracers of $NO_2$, $SO_2$, and CO ($R^2 = 0.47$, 0.36, and 0.49, respectively). CO and $NO_2$ are mainly gases from combustion sources strongly influenced by urban activities such as traffic and domestic heating. Instead, $SO_2$ is generally mainly due to in-

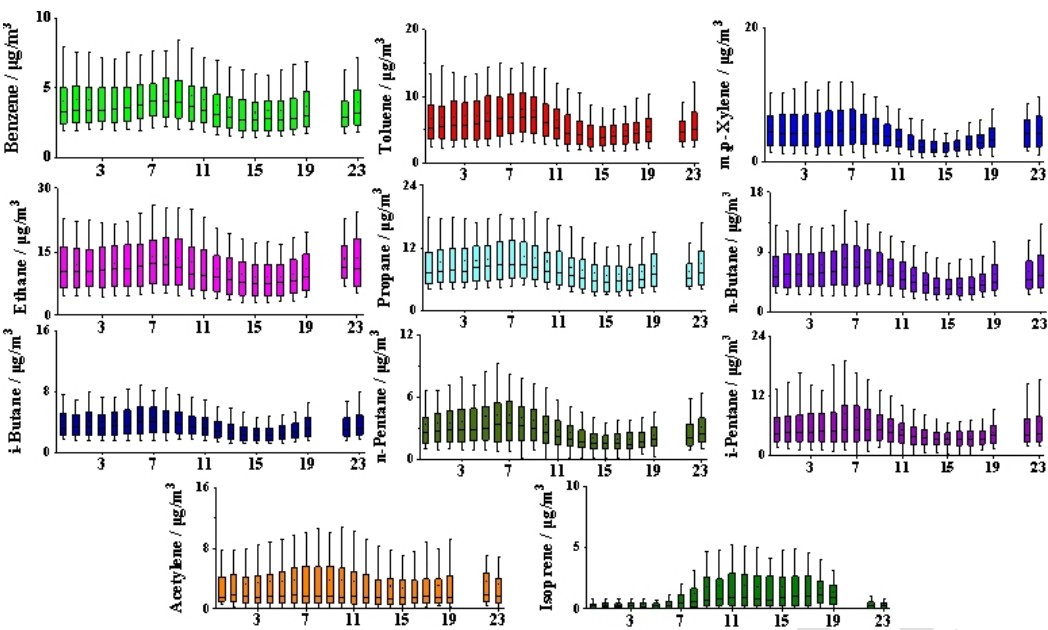

**Figure 3.** Diurnal variations in VOCs compounds measured at Zhengzhou. The upper and lower boundaries of the boxes indicate the 75th and 25th percentiles, respectively; the lines within the boxes mark the median; the whiskers above and below the boxes indicate the 90th and 10th percentiles, respectively.

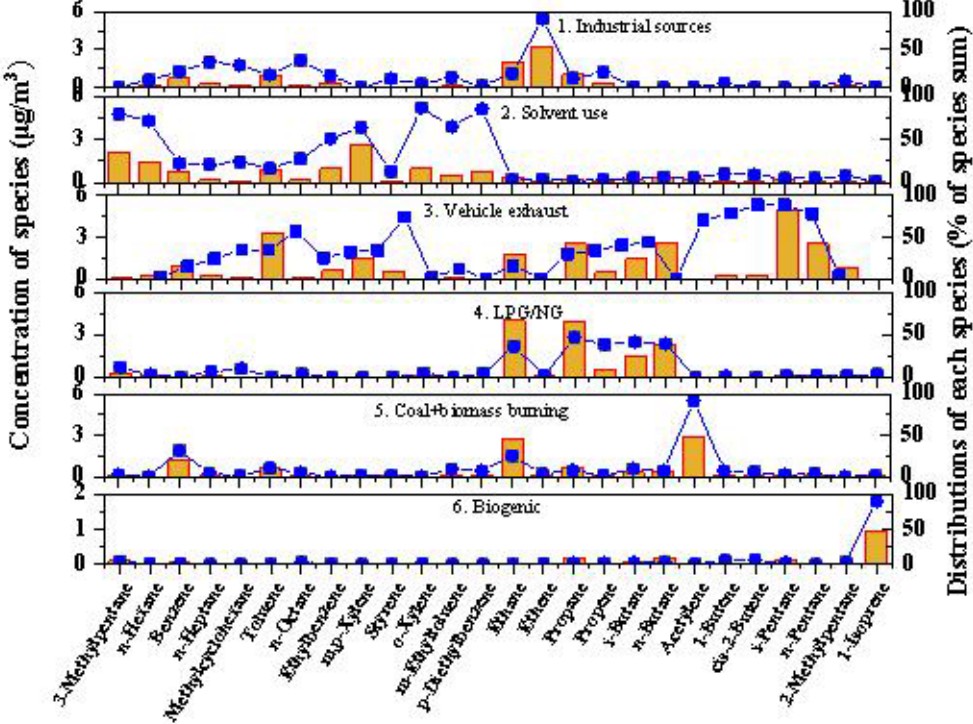

**Figure 4.** Source profiles and contribution percentages during the observation period by PMF model (a bar is a mixing ratio, and a dot is a percentage).

dustrial sources or the combustion of heavy oils and coals. Therefore, source 1 was assigned to industrial sources.

The CPF plots indicated the dominant source directions as northeast and southeast, which correspond to industrial estates (as shown in Fig. 5). Moreover, the highest VOC concentration from industrial sources was observed at 10:00 CST, probably attributable to increased industrial activities during daytime.

### 3.3.2 Factor 2: solvent use

Factor 2 was characterized by high levels of $n$-hexane (60.4 %), $n$-octane (26.3 %), methyl cyclohexane (20.4 %), $o$-xylene (79.6 %), $m/p$-xylene (61.7 %), $p$-diethylbenzene (84.0 %), $o$-xylene (79.0 %), $m$-ethyl toluene (57.8 %), $o$-ethyl toluene (29.2 %), ethylbenzene (51.1 %), and toluene (20.8 %). These compounds are major components emitted through the use of various solvents or industrial processes (Zhou et al., 2019; M. Wang et al., 2021). However, there were almost no tracers of ethane, ethene, acetylene, and benzene related to combustion sources, and this source appeared to exhibit poor correlation with gas tracers ($R^2 < 0.10$). Therefore, source 2 was identified as solvent use.

The CPF plots of this factor suggested southeast was the dominant source direction, possibly reflecting the presence of the large manufacturers of automobiles in the southeastern urban area.

### 3.3.3 Factor 3: vehicle emissions

Factor 3 was identified by high percentages of $C_2$–$C_6$ alkanes (i.e., ethane (37.8 %), propane (46.9 %), isobutane (40.9 %), $n$-butane (38.5 %), $n$-pentane (26.0 %), and $i$-pentane (31.6 %)), $C_2$–$C_4$ alkanes (i.e., ethylene (31.4 %), propylene (38.4 %), and 1-butene (67.4 %)), and $C_6$–$C_8$ aromatics (i.e., benzene (21.4 %), toluene (30.7 %), and $m/p$-xylene (25.6 %)). These components are considered typical products of incomplete combustion processes (Baudic et al., 2016; Gaimoz et al., 2011; Liu et al., 2008; Song et al., 2018). It is reported that $i$-pentane usually originates from gasoline evaporation (Mo et al., 2017) and that 2-methylpentane and 3-methylpentane are tracers of the emissions of gasoline-powered vehicles (Tsai et al., 2003; Liu et al., 2008; Song et al., 2018). The T / B ratio was 2.0 in this profile, which further confirms the effect of vehicular emissions (Yao et al., 2021). Moreover, the source correlated significantly with CO and NO$_2$ ($p < 0.01$) but not with SO$_2$ ($p > 0.05$); therefore, source 3 was identified as vehicle emissions.

Factor 3 showed larger CPF values when the wind came from the north possibly because the site is adjacent to Jinshui Road, which is the main road in Zhengzhou. The diurnal pattern of the traffic source was characterized by two peaks: one in the morning and the other in the evening, consistent with the relative strength of local traffic flow.

### 3.3.4 Factor 4: LPG/NG

Factor 4 was dominated by strong presence of ethane (16.3 %), propane (14.6 %), isobutane (38.9 %), $n$-butane (44.4 %), $n$-pentane (60.3 %), and $i$-pentane (66.2 %), which can also be released via fuel evaporation (gasoline and LPG/NG) (Zhang et al., 2019). Pentanes are the most abundant VOC species associated with gasoline evaporation (Liu et al., 2008; Zhang et al., 2013; Shen et al., 2018), and butanes are reported as tracers of LPG (McCarthy et al., 2013). In particular, the aromatics of this source were very low. Similar to factor 3, the source did not correlate with SO$_2$ ($p > 0.05$) but had positive correlation with both CO and NO$_2$ ($p < 0.01$); therefore, this source was considered as LPG/NG.

The dependence of factor 4 on WD was not as significant as for other sources. It is related to compact residential areas with greater human activity. There are residential areas around the monitoring site, and LPG/NG leakage might have occurred in the process of daily life.

### 3.3.5 Factor 5: fuel burning

Factor 5 was distinguished by substantial amounts of acetylene (72 %), which is a marker of combustion sources (Hui et al., 2021). Additionally, the source was also characterized by considerable amounts of benzene and $C_2$–$C_3$ hydrocarbons, which are representative species of incomplete combustion processes (Zheng et al., 2021). Meanwhile, the independent tracers (i.e., NO$_2$, SO$_2$, and CO) exhibited correlation with this factor ($R^2 > 0.3$, $p < 0.01$); therefore, source 5 was considered to be fuel burning.

Factor 5 displayed high CPF values when the wind was from the east. This is possibly related to the heating companies located within 1.0 km to the east of the site. The diurnal variation in this factor was characterized by an apparent increase at night, which could be related to the accumulation of pollutants associated with nighttime heating.

### 3.3.6 Factor 6: biogenic sources

Factor 6 exhibited a significantly high composition of isoprene, which is mainly produced by vegetation through photosynthesis (C. Song et al., 2019). Accordingly, source 6 was labeled as biogenic sources.

The CPF plots indicated that this factor mainly originated from the west with conditional probability values of approximately 0.44. It is mainly affected by Zijinshan Park, which is located $\sim 1$ km to the west of the monitoring site. Additionally, the diurnal pattern of the biogenic sources showed obvious $T$ dependence, with the highest concentration at midday that could be associated with photosynthetic activity.

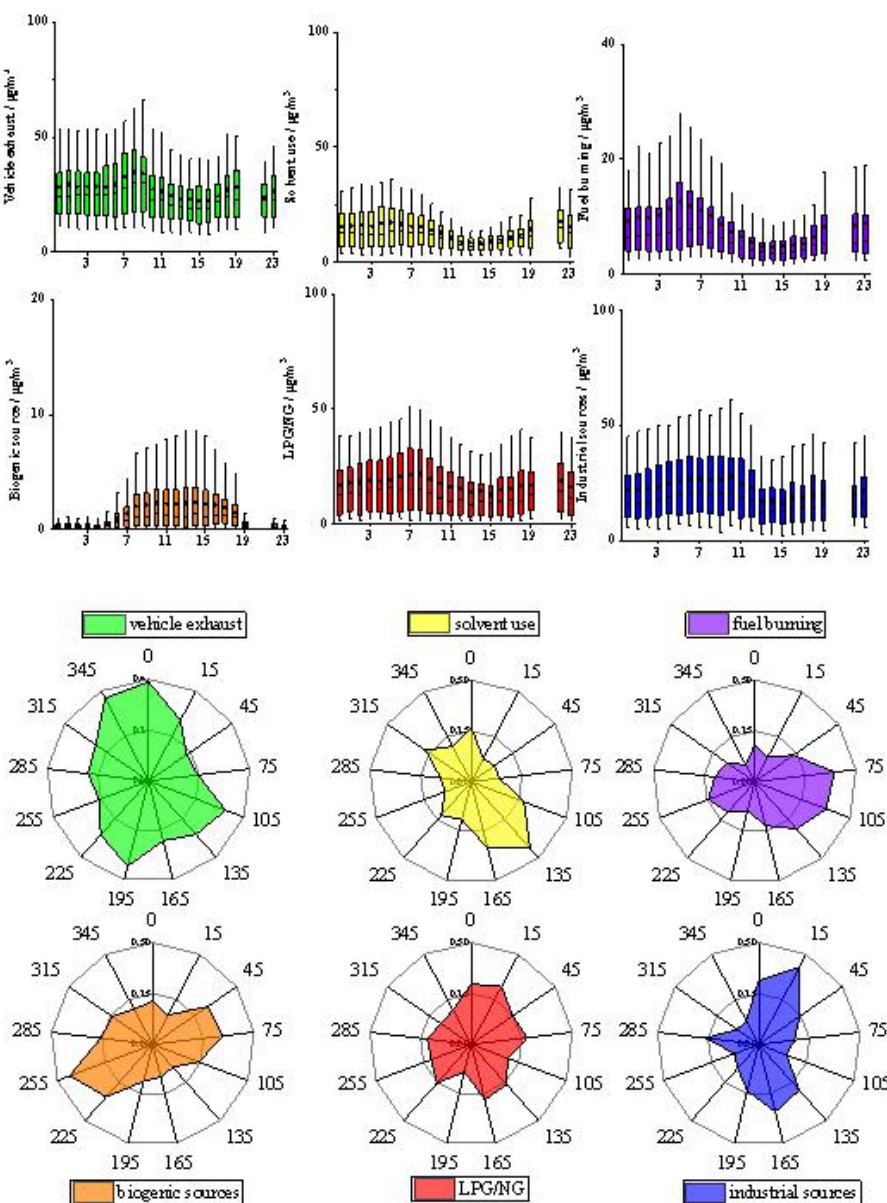

**Figure 5.** Conditional probability function (CPF) plots of local VOC sources in Zhengzhou. The mean (dot), median (horizontal line), 25th and 75th percentiles (lower and upper box), and 10th and 90th percentiles (lower and upper whiskers) for the entire study are shown.

### 3.3.7 Comparison between the composite source profiles and the PMF factors

As shown in Fig. S5, the source profiles derived from the PMF analysis were compared with their sources attributed from the source profiles. The data of the source profiles were derived from a local tunnel experiment and a review of the most recent literature. The source profiles for solvent use correlated most strongly between the two methods ($R = 0.84$). A study by Jin et al. (2021) identified that low-carbon alkanes (e.g., ethane, propane, and isopentane), alkenes (e.g., ethylene, propene, and 1-butene), and aromatics (e.g., benzene, toluene, and $m/p$-xylene) were the main groups in the tunnel

study and that vehicle emissions agreed well between this factor and the source profiles ($R = 0.59$). Different profiles of combustion sources were investigated, and the correlation between the results of NG combustion and the PMF factor was strongest ($R = 0.57$). The above results are to be expected because Zhengzhou has gradually phased out coal-fired boilers and replaced them with gas boilers in recent years. As for industrial sources, the correlation between the two methods was 0.43. The $C_2$–$C_3$ hydrocarbons accounted for a high proportion in factor 1, whereas the content of aromatics was lower. Industrial production of nonmetallic mineral products in Zhengzhou is reasonably well developed,

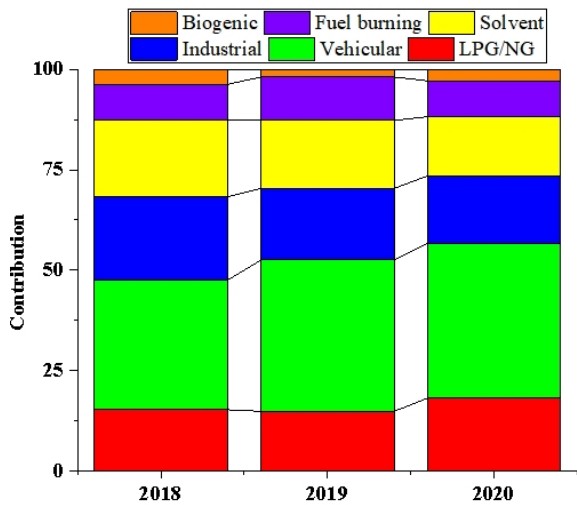

**Figure 6.** The contributions of each VOC source during 2018–2020.

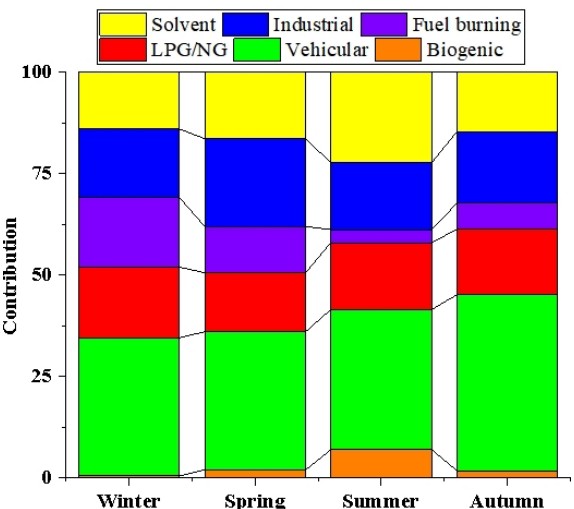

**Figure 7.** Seasonal variation in source contributions to VOC concentration.

which leads to large emissions of ethane and propane. The main contributing source of aromatic hydrocarbons is rubber and plastics. However, the scale of the enterprises involved in their production is far lower than that in the Pearl River Delta; consequently, the emission of aromatic hydrocarbons is lower in Zhengzhou.

## 3.4 VOC source contributions

### 3.4.1 Interannual variation

The concentration contributions of each VOC source during 2018–2020 are shown in Fig. 6. In 2018, vehicular emissions were the largest contributor to VOC mixing ratios (32 %), followed by industrial sources (21 %), solvent use (19 %), and LPG/NG (15 %). The contribution of fuel burning and biogenic sources accounted for 9 % and 4 % of the total VOC concentration, respectively. As for 2019, vehicle emissions made the largest contribution (38 %) to atmospheric VOCs. The second most significant source was industrial sources, accounting for 18 % of the total. The contribution of solvent use, LPG/NG, fuel burning, and biogenic sources to atmospheric VOCs was 17 %, 15 %, 11 %, and 2 %, respectively, in 2019. Vehicular emissions, LPG/NG, industrial sources, solvent use, fuel burning, and biogenic sources accounted for 39 %, 18 %, 17 %, 15 %, 9 %, and 3 %, respectively, in 2020.

In summary, vehicle emissions and industrial sources made the largest contributions in all 3 years. Moreover, the proportions of the contributions of vehicle emissions and LPG/NG have increased with each passing year. As a transportation hub city, the number of motor vehicles in Zhengzhou has maintained a rate of growth of 0.4 million annually during the past 5 years, and the total number of vehicles exceeded 4.5 million in 2020 (X. Gu et al., 2019). Thus, vehicle emissions represent an important source of ambient

VOCs in Zhengzhou. Both LPG and NG are used widely in residential life, industrial production, and motor vehicles. In terms of the actual situation of the monitoring site, residential emissions might represent the main source of LPG/NG. With adjustment of the energy structure, most industrial enterprises and taxis in Zhengzhou now use NG as energy or fuel. It should be noted that the proportion of the contributions from industrial and solvent sources has presented an annual downward trend. In recent years, Zhengzhou has implemented special actions designed to reduce VOC emissions, focusing on the control of industrial and scattered small-scale pollution enterprises. According to the results obtained using the PMF model, it is speculated that the effect of such policy control has been remarkable.

### 3.4.2 Seasonal variation

Owing to different meteorological conditions and emission strengths, the source contributions vary seasonally. As shown in Fig. 7, the seasonal variation in biogenic emissions was substantial, with the highest contribution in summer (7 %) and the lowest contribution in winter (< 1 %). This might reflect both $T$ and UV intensity. Conversely, the contributions from fuel burning were larger in winter and lower in summer. Fuel burning accounted for a larger proportion of emissions in winter (17 %) than in summer (3 %). Meanwhile, industrial emissions contributed a high percentage of VOCs in spring (22 %), whereas the contributions in the other three seasons were comparable. Additionally, motor vehicle emissions showed no obvious seasonal characteristics, and the contribution to atmospheric VOCs in each of the four seasons exceeded 30 % of the total, indicating that motor vehicles have considerable impact on the air quality of Zhengzhou.

### 3.4.3 Comparison with other studies

#### Comparison with emission inventory (EI) studies

This study compared the annual average relative contributions of different sources with results from published emission inventory (EI) studies. In the anthropogenic VOC emission inventories established by Lu et al. (2020), VOC sources were classified into eight categories: stationary combustion, on-road mobile sources, non-road mobile sources, industrial processes, solvent use sources, fuel oil storage and transportation, biomass burning, and others. Vehicle emissions represented the most abundant source of anthropogenic VOCs in both the EI and the PMF analysis, accounting for 29.7 % and 36.3 % of the total, respectively. The differences observed between the EI and PMF results were primarily because most of the vehicles considered were in urban areas. The contributions of industrial sources and of solvent use sources represented the second and third largest, respectively, in both the EI and the PMF results. However, the overall contribution of those two sources obtained from the PMF results was lower than that derived from the EI analysis. The relative contribution of combustion sources resolved from the PMF analysis was higher than that obtained from the EI results, accounting for 9.5 % and 4.3 %, respectively. Such large differences occurred primarily because of uncertainties in the activity data obtained from statistical information. Residential energy consumption and emissions were poorly recorded in comparison with other sources (Chen et al., 2016; Tao et al., 2018), leading to higher uncertainties in the related estimations of such emissions. Thus, it is necessary to estimate the emissions of residential fossil fuel combustion through scientific approaches.

#### Comparison with other PMF studies

The source contributions determined in this research were also compared with the results from other studies. It can be seen from Table 2 that the source apportionment of VOCs in this study was broadly within the values reported for other Chinese cities (Huang et al., 2019; Hui et al., 2018; Q. Li et al., 2020b; Mo et al., 2017; Yan et al., 2017). It was found that traffic emissions represent the main source of VOCs in Zhengzhou and the other five cities, indicating that vehicle emissions have considerable impact on the concentration of VOCs in the urban atmosphere. It should be noted that the contribution of each source determined in this study was very similar to that reported for Wuhan (Hui et al., 2018). Given that Zhengzhou and Wuhan are both important transportation hub cities in central China, this result is in line with expectation.

### 3.5 AOC

The AOC during the sampling periods was quantified, as shown in Fig. 8. The calculated averaged value of total

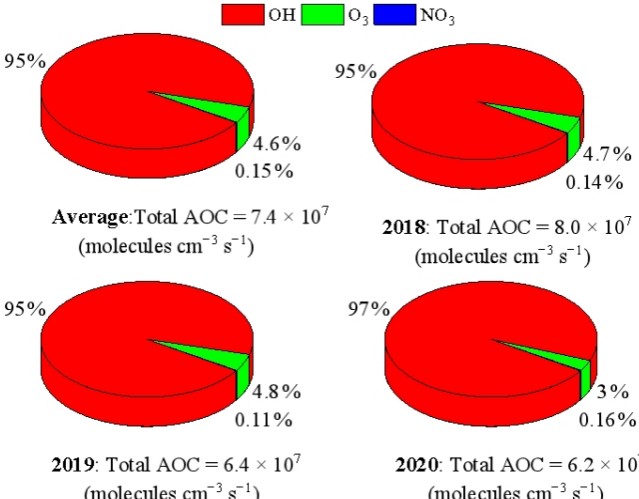

**Figure 8.** Comparison of the relative contributions of OH, $O_3$, and $NO_3$ of the AOC in Zhengzhou during the sampling periods.

AOC was $7.4 \times 10^7$ molec. $cm^{-3}$ $s^{-1}$, comparable with values reported for a suburban site between Beijing and Tianjin (Yang et al., 2020b) but substantially higher than observed in Shanghai (Zhu et al., 2020), Hong Kong (Xue et al., 2016), Chile (Elshorbany et al., 2009), and Berlin (Geyer et al., 2001). Among the AOC categories, OH exhibited the highest average concentration ($7.0 \times 10^7$ molec. $cm^{-3}$ $s^{-1}$), accounting for 95 % of the total AOC, followed by $O_3$ ($3.4 \times 10^6$ molec. $cm^{-3}$ $s^{-1}$) and $NO_3$ ($1.1 \times 10^5$ molec. $cm^{-3}$ $s^{-1}$), which contributed 4 % of the total AOC. Thus, OH is the main contributor of atmospheric oxidation in Zhengzhou, similar to the results reported for other regions by other studies (Yang et al., 2021; Zhu et al., 2020).

During 2018–2020, the total AOC presented a decreasing trend annually (as shown in Fig. 8), with mean values of 8.0, 6.4, and $6.2 \times 10^7$ molec. $cm^{-3}$ $s^{-1}$, respectively. As expected, OH was the predominant oxidant in each of the 3 years, accounting for 95 %, 95 %, and 97 % of the total AOC, respectively. The primary pollutants (e.g., CO, $CH_4$, and VOCs) decreased significantly in 2020 owing to large reductions in economic activity and the associated emissions during the COVID-19 lockdown. However, the average OH concentration was highest in 2020 ($4.8 \times 10^7$ molec. $cm^{-3}$), far higher than that in either of the 2 previous years. Thus, the level of atmospheric oxidation needs additional attention. It should be noted that oxidation of the atmosphere did not decrease or even increase significantly during the pandemic, which has been reported by many previous studies (Y. Wang et al., 2021).

As shown in Fig. S7, the mean AOC values showed pronounced seasonal variation. The highest total AOC value was detected in summer ($7.5 \times 10^7$ molec. $cm^{-3}$ $s^{-1}$), followed by winter ($6.4 \times 10^7$ molec. $cm^{-3}$ $s^{-1}$), spring ($5.8 \times 10^7$ molec. $cm^{-3}$ $s^{-1}$), and autumn

**Table 2.** Comparison of source contributions resolved by PMF models in different cities (%).

| City | Sampling periods | Solvent use | Industrial sources | Vehicle exhaust | Fuel burning | LPG/NG | Biogenic sources |
|------|------------------|-------------|--------------------|-----------------|--------------|--------|------------------|
| Taiwan[a] | January–December 2016 | 29 | 15 | 18 | – | – | 4 |
| Wuhan[b] | 1 September 2016 to 31 August 2017 | | 16 | 24 | 19 | 13 | 2 |
| Shuozhou[c] | March and August 2014 | – | 14 | 21 | 30 | 18 | – |
| Ningbo[d] | December 2012, April 2013, July 2013, and October 2013 | 7 | 50 | 16 | – | 27 | – |
| Beijing[e] | March 2016 to January 2017 | 16 | 10 | 19 | – | 12 | 8 |
| This study | January 2018 to December 2020 | 17 | 18 | 36 | 10 | 16 | 3 |

[a] Huang et al. (2019), [b] Hui et al. (2018), [c] Yan et al. (2017), [d] Mo et al. (2017), [e] Q. Li et al. (2020b).

($5.7 \times 10^7$ molec. cm$^{-3}$ s$^{-1}$). The concentration of OH in summer was significantly higher than that in the other seasons, which can be ascribed to the relatively favorable meteorological conditions. Meanwhile, high concentrations of isoprene were observed in summer, and its high reaction rate coefficients with oxidants (e.g., OH, NO$_3$, and O$_3$) revealed a highly oxidative environment during the summer campaign in Zhengzhou. This seasonal pattern of the AOC is similar to that found in other studies conducted at the national level and in urban and suburban environments (Yang et al., 2021; Q. Li et al., 2020a).

### 3.6 Atmospheric environmental implications

#### 3.6.1 OH reactivity of measured species

The calculated OH reactivity was categorized into SO$_2$, NO$_2$, NO, O$_3$, CO, and VOCs, as shown in Fig. 9. During the sampling period, the average value of total OH reactivity was 45.3 s$^{-1}$. Generally, the OH reactivity assessed in this study was much higher than that determined in Shanghai (Tan et al., 2019; Zhu et al., 2020), Chongqing (Tan et al., 2019), and New York (Ren et al., 2006) but comparable with or lower than that reported in Xianghe (Yang et al., 2020a) and Backgarden (Lou et al., 2010). In Zhengzhou, NO$_2$ made the largest contribution to total OH reactivity (54 %), followed by VOCs (17 %), NO (16 %), CO (11 %), SO$_2$ (3 %), and O$_3$ (1 %). Similar results were reported by previous studies conducted in other regions (Yang et al., 2021). It should be noted that this study calculated only the OH reactivity of the measured species, i.e., the impact of unmeasured species, such as secondary products (oxygenated VOCs and nitrates produced by photochemical reactions) and monoterpenes, were not considered. Previous studies have shown that both undetected primary emissions and unmeasured secondary products could contribute to missing reactivity (Yang et al., 2016). Therefore, the value of OH reactivity determined in this study was underestimated to a certain extent.

OVOCs play an important role in influencing the AOC via OH-initiated degradation. A previous review suggested that missing OH reactivity has often been observed, and the difference between the measured and calculated values was attributed to a lack of measurement data for OVOCs (Yang et al., 2016). Steiner et al. (2008) found that OVOCs accounted for 30 %–50 % of the modeled urban VOC reactivity by using the regional Community Multiscale Air Quality model (CMAQ). In another urban study, OVOCs were found to contribute between 11 % and 24 % during summertime in Houston (Mao et al., 2010). In the CAREBeijing-2006 and PRIDE-PRD campaigns, Lou et al. (2010) and Lu et al. (2013) reported the significant contributions of OVOCs to OH reactivity based on model simulations. Since OVOCs were not simultaneously measured in four Chinese megacities, the OVOC contributions were simulated with a box model. OVOCs explained between 20 % and 23 % of the total OH reactivity (Tan et al., 2019). In summary, OVOCs behaved as a major contributor to the total OH reactivity.

Table S5 shown the OH reactivity towards the total VOCs and the comparison with other studies. The OH reactivity of the total VOCs was 6.7 s$^{-1}$, i.e., much lower than that reported for Heshan (13.6 s$^{-1}$) (Yang et al., 2017) and Beijing (8.3 s$^{-1}$) (Yang et al., 2021) but close to that found in Guangzhou (6.4 s$^{-1}$) and Chongqing (6.8 s$^{-1}$) (Tan et al., 2019) and higher than that determined in Shanghai (3.2 s$^{-1}$) (Tan et al., 2019). The detailed contributions of each VOC group to total OH reactivity are presented in Tables S6 and S7. The contribution of alkenes to OH reactivity was predominant, accounting for 5.2 s$^{-1}$ of the total OH reactivity of VOCs. During the sampling period, isoprene made the largest contribution to the OH reactivity of the total VOCs, followed by ethene, $m/p$-xylene, propene, styrene, $cis$-2-butene, $trans$-2-butene, toluene, $i$-pentane, and $trans$-2-pentene, which collectively accounted for 70 % of the OH reactivity of the total VOCs.

As shown in Fig. 9, OH reactivity showed substantial interannual and seasonal variations. The statistical results exhibit a decreasing trend of OH reactivity during 2018–2020 in Zhengzhou with mean values of 50.2, 46.9, and 36.9 s$^{-1}$, respectively. This trend might reflect the implementation of emission reduction measures such as traffic-related measures and the "Coal to Gas" project. The OH reactivity in 2020 was 21 % lower than that in the previous year, which is closely related to the emission reduction associated with the COVID-

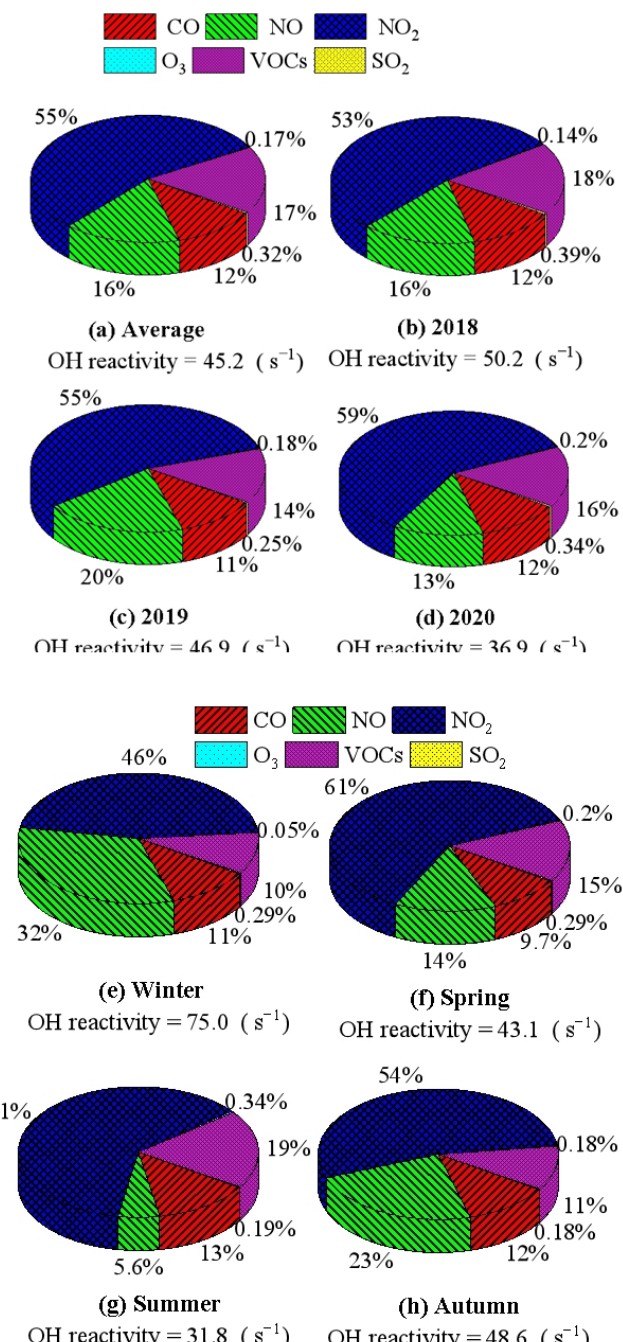

**Figure 9.** Comparison of the relative contributions of the OH reactivity in Zhengzhou.

19 pandemic. Seasonally, the mean value of OH reactivity decreased in the following order: winter ($74.5\,\mathrm{s^{-1}}$) > autumn ($48.6\,\mathrm{s^{-1}}$) > spring ($43.1\,\mathrm{s^{-1}}$) > summer ($31.8\,\mathrm{s^{-1}}$). The notable differences might be attributable to higher loadings of reactive trace gases, especially $NO_x$ and CO. As a northern city, coal combustion in Zhengzhou during the heating season produces higher concentrations of $NO_x$ and CO, leading to higher OH reactivity. Therefore, we should continu-

ally strengthen policies to control trace gases, especially in autumn and winter. The OH reactivity of the total VOCs was similar with the higher value in winter ($7.3\,\mathrm{s^{-1}}$) and the lower value in autumn ($5.3\,\mathrm{s^{-1}}$). However, the concentrations of the key species were markedly different among the four seasons. Ethylene and propylene had the highest OH reactivity in winter, which is speculated to be related to the emission of combustion sources. Additionally, isoprene made the largest contribution to the total OH reactivity in both summer and spring, reflecting the substantial effect of biogenic sources. Overall, the research on specific OH reactivity clearly elucidated the seasonal and annual variations in the major reactants. Therefore, control strategies based on OH reactivity should focus on the key species.

### 3.6.2 Effect of VOCs on $O_3$ formation

Because VOCs are important precursors of $O_3$ formation in the ground-level atmosphere, it is necessary to adequately estimate the contribution of each VOC species to $O_3$ formation. The concentration contributions of the four VOC categories expressed on different scales from 2018 to 2020 are shown in Fig. S8. The result suggests that aromatics made the largest contribution to the MIR concentration, accounting for a combined ratio of 81 %. Meanwhile, alkenes were the largest contributors to the PE concentration (87 %). Although the concentrations of aromatics and alkenes are relatively low, these two VOC groups play an important role in $O_3$ formation, which is a result supported by many previous studies (Hui et al., 2021; Y. Li et al., 2020). It should be noted that the effect of VOCs on $O_3$ formation was calculated from the sum of measured species and does not involve species that were not measured, such as OVOCs. Therefore, we provided a lower limit of the effect of VOCs on $O_3$ formation in this study.

Among the top 10 reactive species contributing to the PE and MIR weighted concentrations (as shown in Fig. 10), 8 compounds were the same, differing only in their rank order. The top 10 VOCs obtained from the PE and MIR methods represent 67 % and 74 % of VOCs, respectively. Considering kinetic activity, isoprene ranked first (8.8 ppb C) with the PE method, accounting for approximately 20 % of the total PE concentration, and $m/p$-xylene, styrene, ethene, and toluene ranked second to fifth, explaining 16 %, 10 %, 7 %, and 5 % of the total OFP, respectively. In comparison, ethene, $m/p$-xylene, toluene, propane, and isoprene had the highest MIR concentrations, which accounted for approximately 23 %, 21 %, 12 %, 7 %, and 7 %, respectively. The results of this study highlight the contributions of isoprene, ethene, $m/p$-xylene, and toluene to $O_3$ formation, and these species are derived primarily from vehicle emissions and industrial coatings (Xiong and Du, 2020).

On the basis of the results of source apportionment, the source contributions to OFP and PE were calculated (shown in Fig. S9). Vehicle emissions made the greatest contribution

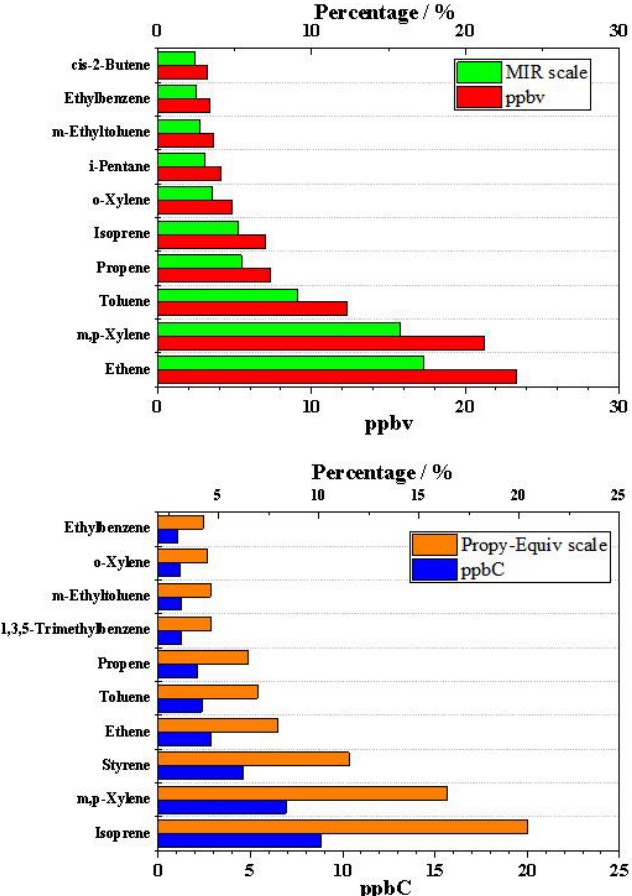

**Figure 10.** Top 10 VOC species that contributed most to the PE and MIR weighted concentrations in Zhengzhou.

to O$_3$ formation (OFP: 23 %; PE: 29 %), followed by solvent use (OFP: 25 %; PE: 24 %). However, the contribution of biogenic emissions accounting for 22 % of the total PE cannot be ignored. Because of the discrepancy between the MIR weighted and PE weighted concentrations, the contribution of this source to OFP was relatively low, accounting for only 6 %. The PE concentration method considers only the kinetic reactivity of VOC species and ignores mechanism reactivity, whereas the MIR method considers the impact of the VOC/NO$_x$ ratio on O$_3$ formation.

## 4   Conclusions

In this study, hourly observational data of 57 VOC species were collected during 2018–2020 at an urban site in Zhengzhou (China). The results showed that the average total VOC mixing ratio was $94.3 \pm 53.1 \, \mu g \, m^{-3}$ ($38.2 \pm 15.6$ ppbv) and that the VOC concentrations were dominated by alkanes in each of the 3 years. During the sampling period, the interannual variation in VOCs gradually reduced as follows: $113.2 \pm 65.2 \, \mu g \, m^{-3}$ in 2018, $90.7 \pm 52.5 \, \mu g \, m^{-3}$ in

2019, and $79.1 \pm 41.7 \, \mu g \, m^{-3}$ in 2020. The VOCs showed clear seasonal dependence with the highest value in winter ($116.5 \, \mu g \, m^{-3}$) and lowest value in summer ($74.2 \, \mu g \, m^{-3}$). The PMF method was used to identify six sources: vehicle emissions (36 %), solvent use (17 %), LPG/NG (16 %), industrial sources (18 %), fuel burning (10 %), and biogenic sources (3 %). The proportion of both vehicle emissions and LPG/NG has increased with each passing year. However, the proportion of industrial and solvent sources presented a decreasing trend. In addition to substantial interannual variation, the VOC sources also showed marked seasonal differences. The contribution of vehicle emissions to the atmosphere in each of the four seasons was > 30 %. Atmospheric VOCs are affected substantially by fuel burning (17 %) in winter; however, the influence of biogenic sources cannot be ignored (7 %) in summer.

This study also focused on the atmospheric environmental implications of VOCs, including the AOC, OH reactivity, and OFP. During the sampling periods, the campaign-averaged value of the total AOC was $7.4 \times 10^7 \, molec. \, cm^{-3} \, s^{-1}$, and OH exhibited the highest average concentration, accounting for 95 % of the total AOC. The average value of the total OH reactivity was $45.3 \, s^{-1}$, and NO$_2$ made the largest contribution to the total OH reactivity (54 %), followed in descending order by VOCs (17 %), NO (16 %), CO (11 %), SO$_2$ (3 %), and O$_3$ (1 %). Investigation of the effects of VOCs on O$_3$ formation revealed that despite the relatively low concentrations of aromatics and alkenes, they played important roles in O$_3$ formation. Ethene, $m/p$-xylene, and toluene contributed substantially to O$_3$ formation in Zhengzhou. The source apportionment results indicate that vehicle emissions and solvent use remain the key sources of the VOCs that contribute to O$_3$ formation.

Overall, investigation of the concentrations, source apportionment, and atmospheric environmental implications clearly elucidated the differences in the major reactants observed in different seasons and years. Therefore, control strategies should consider seasonal and interannual variations when focusing on the key species and sources. The results of this study could support local governments in developing strategies to control VOCs during O$_3$ pollution events.

**Data availability.** All the data presented in this article can be accessed through https://doi.org/10.5281/zenodo.6815259 (Yu, 2022).

**Supplement.** The supplement related to this article is available online at: https://doi.org/10.5194/acp-22-1-2022-supplement.

**Author contributions.** SY and RZ planned and organized the study and were deeply involved in writing the manuscript. SW, RX, and DZ performed the atmospheric measurements and data analysis and wrote the manuscript. MZ and LW assisted heavily with the

atmospheric measurements and data analysis. FS, XuL, and XiL conducted the model development and data analysis.

All coauthors provided useful insights in data analysis and contributed to the writing of the manuscript.

**Competing interests.** The contact author has declared that none of the authors has any competing interests.

**Disclaimer.** Publisher's note: Copernicus Publications remains neutral with regard to jurisdictional claims in published maps and institutional affiliations.

**Acknowledgements.** This work was supported by the Study of Collaborative Control of $PM_{2.5}$ and $O_3$ Pollution in Zhengzhou City (no. 20200321A) and the National Key Research and Development Program of China (no. 2017YFC0212403).

We thank James Buxton from Liwen Bianji (Edanz) (https://www.liwenbianji.cn/, last access: 7 May 2022) for editing the English text of a draft of the manuscript.

**Financial support.** This research has been supported by the National Key Research and Development Program of China (grant no. 2017YFC0212403).

**Review statement.** This paper was edited by Andrea Pozzer and reviewed by two anonymous referees.

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

## Remarks from the typesetter