# Peer review of "Measurement report: Intra-, inter-annual variability and"

_Atmospheric Chemistry and Physics, 2021_

## Author Comment (AC1)

**General comment**

The paper reports a discussion of a measurement dataset of VOCs collected in Zhengzhou (China) between 2018 and 2020. Discussion on trends, potential sources is included in the paper. The approach is not particularly new, however, the dataset and the analysis is quite complete and I believe that the paper could be interesting for the scientific community and suitable for the Journal. However, a few aspects are not completely clear or discussed in sufficient details so that a revision would likely improve the paper, see my specific comments.

**Response:** Thank you for your careful reading of our paper and the valuable comments and constructive suggestions. Below are the point-to-point responses to all the comments (The comments are marked in black font and the responses are marked in dark blue font). The major changes that have been made according to these responses were marked in yellow color in the highlighted copy of the revised manuscript. And our own minor changes were marked in red font. Note that the following line numbers are shown in the corrected version.

**Specific comments**

Lines 64-66. Here it would be better to use some references, especially for CMB applied to gaseous VOCs. I am quite aware of use of CMB receptor model for particulate matter and several source profiles are available in the scientific literature but, likely, much less information is available for source profile of VOCs.

**Response:** Thank you for your suggestions. The references have been supplemented.

(Hellén et al., 2003;Plaisance et al., 2017)

Hellén, H., Hakola, H., Aurila, T., 2003. Determination of source contributions ofNMHCs in Helsinki (60 N, 25 E) using chemical mass balance and the Unmixmultivariate receptor models. Atmos. Environ. 37, 1413–1424.

Plaisance, H., Mocho, P., Sauvat, N., Vignau-Laulhere, J., Raulin, K., Desauziers, V., 2017. Using the chemical mass balance model to estimate VOC source contributions in newly built timber frame houses: a case study. Environ. Sci. Pollut. R. 24, 24156–24166.

Section 2.2 is quite stingy of details and should be enriched. I would suggest to mention the work of Belis et al. (Atmospheric Environment X, 5, 2020, 100053) regarding performances of receptor models and    mention if specific constraints were used in the PMF run and how measurement uncertainties were taken into account and what is the total variable used.

**Response:** Thanks for your suggestions. We have corrected it. The description of PMF has been updated.

"In this study, analysis of the source of the VOCs was performed using the EPA PMF 5.0 model, which is a receptor model used widely for source apportionment (Gao et al., 2018; Yadav et al., 2019). Detailed information regarding this method is available in the user manual (Norris et al., 2014) and other related literature (Song et al., 2019a, 2019b). Two input files are required for PMF: the concentration values and the uncertainty values of the individual VOC species. The uncertainty is calculated using Eq. (1) when the species concentration value is higher than its method detection limit (MDL), or using Eq. (2) when the concentration is less than or equal to the MDL:

$$\mathrm{Unc} = \sqrt{(\mathrm{EF} \times c)^2 + (0.5 \times \mathrm{MDL})^2}, \quad (1)$$

$$\mathrm{Unc} = \frac{5}{6} \times \mathrm{MDL}, \qquad\qquad (2)$$

where $c$ is the concentration of the individual VOC species, and EF is the error fraction, which was set to 10% of the VOC concentration (Yuan et al., 2012).

Owing to the complexity of the chemical reactions, not all of the VOC species were used in the PMF analysis. Based on previous work, this study adopted the following principles for selection of the VOC species. (1) Species with more than

25% of data missing or below the MDLs were rejected, which follows the methodology of previous studies (Zhou et al., 2019). (2) Species with short atmospheric lifetimes were excluded because they rapidly react away in the atmosphere. (3) Species that represent source tracers of emission sources were retained (e.g., in the case of isoprene). Eventually, 27 VOC species were selected for source apportionment analysis. VOC species were grouped into strong, weak and bad according to their signal/noise ratio (S/N), and there were 23 and 4 species grouped into strong and weak, respectively. It should be noted that the volumetric concentration (ppbv) of the individual VOC species was converted to mass concentration ($\mu g\ m^{-3}$) before being input into the PMF model.

Choosing the optimal number of factors in the model is important. The number of factors depends on Q (ture)/Q (robust) and Q/Qexpected (Qexp). In theory, Q (ture)/Q (robust) < 1.5 and a value close to 1 is considered reasonable (Ulbrich et al., 2009), and the rate of change of Q/Qexp should be stable and the ratio should be close to 1 (Baudic et al., 2016; Hui et al., 2019). In this study, the numbers of factors used for the PMF analysis were tested from three to eight, and the optimum six-factor solution with Q/Qexp = 0.94, (Q (ture)/Q (robust) = 1.0) was selected. Additionally, Fpeak values from −1 to 1 with 0.1 intervals were used in the model, and Fpeak = -0.2 was established as the best solution (as shown in Fig. S1)."

In supplementary material, and related to the previous point. It is mentioned principal component analysis on this dataset but there is not trace of it in the paper. In addition, it should be explained how the number of factors was chosen because Figure 1 with a constantly decreasing Q/Qe does not seem to allow this identification by itself.

 **Response:** Sorry for the mistake. We have corrected it. The description has been corrected to "Choosing the optimal number of factors in the model is important. The number of factors depends on Q (ture)/Q (robust) and Q/Qexpected (Qexp). In theory, Q (ture)/Q (robust) < 1.5 and a value close to 1 is considered reasonable (Ulbrich et

al., 2009), and the rate of change of Q/Qexp should be stable and the ratio should be close to 1 (Baudic et al., 2016; Hui et al., 2019). In this study, the numbers of factors used for the PMF analysis were tested from three to eight, and the optimum six-factor solution with Q/Qexp = 0.94, (Q (ture)/Q (robust) = 1.0) was selected. Additionally, Fpeak values from −1 to 1 with 0.1 intervals were used in the model, and Fpeak = -0.2 was established as the best solution (as shown in Fig. S1)".

[Figure]

[Figure]

**Fig. S1** The Q/Qexp and Q (ture)/Q (robust) ratios in different solutions (a); the

Q/Qexp ratio for different Fpeak value solutions (b).

Lines 215-218. It should be mentioned is the differences in these yearly averages are statistically significant considering the large standard deviations (are STD reported as errors?) indicated.

**Response:** Thanks for your suggestions. We have corrected it. The description of Line 256-260 has been corrected to "The interannual variation of the VOCs declined gradually as follows: 113.2±65.2 μg/m3 in 2018, 90.7±52.5 μg/m3 in 2019, and 79.1±41.7 μg/m3 in 2020. It should be mentioned is the differences in these yearly averages are statistically significant considering the large standard deviations indicated."

Lines 333-335. To better explain this reasoning, it should be mentioned that CO and NO2 are mainly gases from combustions sources strongly influenced by urban activities such as traffic and domestic heating. Instead, SO2 is generally mainly due to industrial sources or combustion of heavy oils such as fuels used in ships.

 **Response:** Thanks for your suggestions. We have corrected it. The description of Line 368-372 has been corrected to "Meanwhile, this source had correlation with gas tracers of $NO_2$, $SO_2$, and CO ($R^2$ = 0.42, 0.37, and 0.44, respectively). CO and NO2 are mainly gases from combustions sources strongly influenced by urban activities such as traffic and domestic heating. Instead, SO2 is generally mainly due to industrial sources or combustion of heavy oils and coals. Therefore, source 1 was assigned to industrial sources."

Section 3.3.1. This part could be made more strong if related to the diagnostic ratios. For example, the B/T ratio in the different profiles are similar to those found in literature for the specific sources as discussed previously. Actually, the figure 4 is very small and I do not see clearly. I also suggest to increase the size of this figure.

**Response:** Thanks for your suggestions. Section 3.1 (Source identification) has been rewritten. Meanwhile, Fig. 4 has been drawn at the same time.

[Figure]

**Fig.4** Source profiles and contribution percentages during the observation period by PMF model (bar is a mixing ratio and dot is a percentage).

Section 3.3.2. At the end it is not clear if the trends are present and statistically significant. Actually in Figure 5 it seems that rends are not so relevant in relative terms.

**Response:** Thanks for your suggestions. The concentration contributions of each VOC source during 2018–2020 were updated (as shown in Fig.5). The proportion of vehicle emissions and LPG/NG has increased with each passing year. And the proportion of industrial and solvent sources presented an annual down trend.

[Figure]

**Fig. 5** The contributions of each VOC source during 2018-2020.

Line 485. Why here it is mentioned ppbv rather than s-1?

**Response:** Sorry for the mistake. We have corrected it.

Figures 2 and 3. I suggest to change the vertical scale to maximize the visibility of the data. For example, B is always less than 3 in Figure 3, so why to choose a scale at 6 that compress everything? In addition, in the data in Figure 3 it is missing the results for 20 and 21 (i.e. 8 and 9 pm). The same problems are also present in Fig. 7, Fig. S3.

**Response:** Sorry for the mistake. We have corrected it. Figures 2 and 3 have been updated. Meanwhile, the standard gas calibrated instrument at 20:00 and 21:00 every day, so there is no data during this period.

[Figure]

**Fig. 2** Monthly changes in the concentrations of the typical VOCs species in Zhengzhou. The upper and lower boundaries of the boxes indicate the 75th and 25th percentiles, respectively; the lines within the boxes mark the median; the whiskers above and below the boxes indicate the 90th and 10th percentiles, respectively.

[Figure]

**Fig. 3** Diurnal variations in VOCs compounds measured at Zhengzhou. The upper and lower boundaries of the boxes indicate the 75th and 25th percentiles, respectively; the lines within the boxes mark the median; the whiskers above and below the boxes indicate the 90th and 10th percentiles, respectively.

Tables S2, S3, and S4. Better to indicate the measurement units and also explain what is Pr.

**Response:** Thanks for your suggestions. Tables S4-S7 were indicated the measurement units. And Pr means precipitation.

**Table S4** Variations in the monthly average of meteorological parameters (T, RH, UV, and WS) and pollutant gases ($O_3$, $NO_2$, CO, and TVOC).

| Month | RH (%) | Pr (mm) | T (℃) | WS (m/s) | UV (W/m$^2$) | TVOC (µg/m$^3$) | $NO_2$ (µg/m$^3$) | $O_3$ (µg/m$^3$) | CO (mg/m$^3$) |
|---|---|---|---|---|---|---|---|---|---|
| 1 | 46.4±22 | 7±0.1 | 3.1±3.3 | 1.3±0.8 | 85±33.8 | 145±80.7 | 65.4±29.4 | 22.6±19.5 | 1.4±0.7 |
| 2 | 55.9±16.1 | 3.9±0 | 4.3±5 | 1.2±1 | 114.5±45.3 | 99.8±62.4 | 45.3±29.5 | 50.4±35.5 | 1.2±0.6 |
| 3 | 38.1±16.7 | 3.3±0.1 | 13.9±4.8 | 1.6±0.8 | 206.5±55 | 91.6±44.7 | 48.4±29.3 | 65.2±42.8 | 0.5±0.3 |
| 4 | 52.2±19.7 | 20.2±0.2 | 16.7±5.3 | 1.9±1.1 | 238.5±86.6 | 95.4±37.8 | 41.4±23.6 | 76.7±49.1 | 0.8±0.4 |
| 5 | 40.9±17.6 | 0.2±0 | 24.8±5.5 | 1.5±0.9 | 315.2±62.6 | 72.2±35.8 | 39±29.4 | 100.1±62.1 | 0.7±0.3 |
| 6 | 48.4±22.3 | 20.6±0.2 | 30±7.6 | 0.6±0.4 | 291.6±112 | 76.1±32.6 | 32.9±23.1 | 114±63.9 | 0.6±0.3 |
| 7 | 60±15.3 | 18.2±0.3 | 30.6±4.6 | 0.4±0.2 | 305.6±70.3 | 81.3±38.3 | 36.4±27.8 | 110.4±66.9 | 0.7±0.3 |
| 8 | 70±18 | 60.5±0.6 | 27.8±3.7 | 0.4±0.2 | 265±80.1 | 65.3±25.1 | 32.6±20.6 | 95.5±58.5 | 0.8±0.3 |
| 9 | 65±19.2 | 1.9±0 | 23.8±4.3 | 1.2±0.8 | 208.3±85.9 | 80.6±39.3 | 45.7±35.8 | 95±73.3 | 0.9±0.4 |
| 10 | 63±21.5 | 81.8±0.5 | 16.8±4.8 | 1.5±1.3 | 160±61.8 | 86.5±56 | 49.3±30 | 54.6±49.7 | 0.9±0.5 |
| 11 | 54.9±22.4 | 4±0.1 | 11.3±5.3 | 1.6±1.3 | 118.7±42.3 | 91.2±51.6 | 55.4±30 | 35±31.6 | 1±0.5 |
| 12 | 58.5±23.8 | 3.4±0 | 5.6±3.9 | 1.4±0.8 | 102.9±41.7 | 104.5±55.2 | 49.8±24.5 | 28.2±25 | 1.2±0.6 |

**Table S5** The OH reactivity towards the total VOCs and the comparison with other studies (unit: s$^{-1}$).

| | The OH reactivity of the total VOCs | The OH reactivity of the total OVOCs | The OH reactivity after deducting OVOCs | References |
|---|---|---|---|---|
| Zhengzhou | 6.7 | - | 6.7 | This study |
| Xianghe | 7.9 | 2.4 | 5.5 | Yang et al., 2020 |
| Beijing | 15.5 | 7.2 | 8.3 | Yang et al., 2021 |
| Heshan | 18.3 | 4.7 | 13.6 | Yang et al., 2017 |
| Shanghai | 6.21 | 2.97 | 3.24 | Tan et al., 2019 |
| Guangzhou | 10.9 | 4.6 | 6.4 | Tan et al., 2019 |
| Chongqing | 8.9 | 2.136 | 6.8 | Tan et al., 2019 |

**Table S6** The detailed contribution of each VOC group to the total OH reactivity during the sampling periods (unit: $s^{-1}$).

| Species | 2018 | Species | 2019 | Species | 2020 | Species | Average |
|---|---|---|---|---|---|---|---|
| Isoprene | 1.7 | Isoprene | 1.1 | Isoprene | 1.1 | Isoprene | 1.8 |
| Ethene | 1.4 | Ethene | 0.8 | Ethene | 0.9 | Ethene | 1.1 |
| cis-2-Butene | 0.6 | Propene | 0.6 | m/p-Xylene | 0.5 | Propene | 0.5 |
| m/p-Xylene | 0.6 | m/p-Xylene | 0.5 | Propene | 0.4 | m/p-Xylene | 0.5 |
| Propene | 0.6 | Styrene | 0.3 | Styrene | 0.4 | Styrene | 0.4 |
| Styrene | 0.6 | trans-2-Butene | 0.3 | Toluene | 0.2 | cis-2-Butene | 0.3 |
| trans-2-Butene | 0.4 | cis-2-Butene | 0.2 | trans-2-Butene | 0.2 | trans-2-Butene | 0.3 |
| Toluene | 0.3 | Toluene | 0.2 | cis-2-Butene | 0.2 | Toluene | 0.3 |
| i-Pentane | 0.2 | i-Pentane | 0.2 | 1-Butene | 0.1 | i-Pentane | 0.2 |
| n-Pentane | 0.2 | 1-Butene | 0.2 | i-Pentane | 0.1 | 1-Butene | 0.2 |
| Cyclopentane | 0.2 | trans-2-Pentene | 0.2 | n-Butane | 0.1 | n-Butane | 0.1 |
| 1-Hexene | 0.2 | n-Butane | 0.2 | Propane | 0.1 | trans-2-Pentene | 0.1 |
| 1,3,5-Trimethylbenzene | 0.2 | Propane | 0.1 | trans-2-Pentene | 0.1 | Propane | 0.1 |
| cis-2-Pentene | 0.2 | n-Pentane | 0.1 | o-Xylene | 0.1 | n-Pentane | 0.1 |
| trans-2-Pentene | 0.2 | o-Xylene | 0.1 | 1,2,4-Trimethylbenzene | 0.1 | o-Xylene | 0.1 |
| n-Butane | 0.2 | i-Butane | 0.1 | Ethylbenzene | 0.1 | 1,3,5-Trimethylbenzene | 0.1 |
| 1-Butene | 0.1 | Ethylbenzene | 0.1 | i-Butane | 0.1 | cis-2-Pentene | 0.1 |
| o-Xylene | 0.1 | 3-Methylpentane | 0.1 | 1,3,5-Trimethylbenzene | 0.1 | Ethylbenzene | 0.1 |
| Propane | 0.1 | n-Hexane | 0.1 | n-Pentane | 0.1 | 1-Hexene | 0.1 |
| Ethylbenzene | 0.1 | cis-2-Pentene | 0.1 | 1,2,3-Trimethylbenzene | 0.1 | Cyclopentane | 0.1 |

**Table S7** The detailed contribution of each VOC group to the total OH reactivity in different seasons (unit: $s^{-1}$).

| Species | Winter | Species | Spring | Species | Summer | Species | Autumn |
|---|---|---|---|---|---|---|---|
| Ethene | 1.3 | Isoprene | 1.2 | Isoprene | 2 | Isoprene | 0.6 |
| Propene | 1 | Ethene | 1.1 | m/p-Xylene | 0.5 | Propene | 0.6 |
| m/p-Xylene | 0.6 | Propene | 0.5 | Propene | 0.4 | m/p-Xylene | 0.5 |
| Isoprene | 0.5 | trans-2-Butene | 0.5 | Styrene | 0.3 | Styrene | 0.4 |
| Styrene | 0.4 | m/p-Xylene | 0.4 | Ethene | 0.2 | Ethene | 0.4 |
| cis-2-Butene | 0.3 | cis-2-Butene | 0.3 | cis-2-Butene | 0.2 | Toluene | 0.3 |
| trans-2-Butene | 0.3 | Styrene | 0.2 | Toluene | 0.2 | trans-2-Butene | 0.2 |
| Toluene | 0.3 | 1-Butene | 0.2 | trans-2-Butene | 0.2 | n-Butane | 0.2 |
| 1-Butene | 0.3 | Toluene | 0.2 | i-Pentane | 0.2 | Propane | 0.2 |
| i-Pentane | 0.2 | trans-2-Pentene | 0.2 | 1-Butene | 0.2 | i-Pentane | 0.2 |
| trans-2-Pentene | 0.2 | i-Pentane | 0.2 | trans-2-Pentene | 0.1 | 1-Butene | 0.2 |
| Propane | 0.2 | Acetylene | 0.1 | n-Butane | 0.1 | cis-2-Butene | 0.1 |
| n-Butane | 0.2 | n-Butane | 0.1 | 3-Methylpentane | 0.1 | trans-2-Pentene | 0.1 |
| n-Pentane | 0.1 | Propane | 0.1 | Ethylbenzene | 0.1 | 1,3,5-Trimethylbenzene | 0.1 |
| cis-2-Pentene | 0.1 | o-Xylene | 0.1 | o-Xylene | 0.1 | n-Pentane | 0.1 |
| i-Butane | 0.1 | n-Pentane | 0.1 | Propane | 0.1 | i-Butane | 0.1 |
| n-Hexane | 0.1 | 3-Methylpentane | 0.1 | m-Ethyltoluene | 0.1 | o-Xylene | 0.1 |
| o-Xylene | 0.1 | cis-2-Pentene | 0.1 | n-Hexane | 0.1 | Ethylbenzene | 0.1 |
| Ethane | 0.1 | n-Hexane | 0.1 | n-Pentane | 0.1 | Cyclopentane | 0.1 |
| Acetylene | 0.1 | i-Butane | 0.1 | cis-2-Pentene | 0.1 | 3-Methylpentane | 0.1 |

Figure S2. Please correct Mixing on the y-axis label.

**Response:** Sorry for the mistake. Figures S2 have been updated.

[Figure]

**Fig. S2** Monthly changes in the concentrations of VOCs in Zhengzhou.

Line 56. Better contributors to.

**Response:** Thanks for your suggestions.. We have corrected it. The description of Line 57-60 has been corrected to "In many regions, alkanes represent the dominant VOC species, while studies which do not report OVOCs usually identify aromatics and alkenes as better contributors of ozone formation potential (OFP) (Li et al., 2019b; Yan et al., 2017)."

Line 73. Better hot topic.

**Response:** Thanks for your suggestions.. We have corrected it. The description of Line 74-76 has been corrected to "In addition to the study of VOC characteristics and source apportionment, analysis of atmospheric oxidation characteristics is another area of hot topic. "

Line 415. Probably it is night time.

**Response:** Thanks for your suggestions.. We have corrected it.The description of Line 435-437 has been corrected to "The diurnal variation of this factor was characterized by apparent increase at night, which could be related to the accumulation of pollutants associated with nighttime heating."

Line 477. Remove the t in excess.

**Response:** Sorry for our carelessness. We have corrected it.

---

## Author Comment (AC2)

General Comments:

The study entitled "Measurement report: Intra-, inter-1 annual variability and source apportionment of VOCs during 2018-2020 in Zhengzhou, Central China" by Yu et al. contains some interesting long-term data and falls within the scope of ACP. Unfortunately, the work is currently not of sufficient quality to be published in ACP. The work may become suitable after major revisions.

**Response:** Thank you for your careful reading of our paper and the valuable comments and constructive suggestions. Below are the point-to-point responses to all the comments (The comments are marked in black font and the responses are marked in dark blue font). The major changes that have been made according to these responses were marked in yellow color in the highlighted copy of the revised manuscript. And our own minor changes were marked in red font. Note that the following line numbers are shown in the corrected version.

Major concerns:

①One of my major concerns is that the authors making big statements about relative VOC abundances, the contribution ozone formation potential and the contribution to the total OH reactivity declaring certain compounds to be the "most abundant" VOCs without having measured oxygenated VOCs. Certain oxygenated compounds such as methanol, acetone or formaldehyde can contribute significantly to the total VOC burden, while formaldehyde and acetaldehyde often contribute significantly to the total OH reactivity. Since OVOC mixing ratios in China can be higher than alkane mixing ratios see e.g. Sun et al. 2019 Environ Sci Pollut Res (2019) 26:27769–27782 a lot of statements need to be qualified to state that they apply only to studies which failed to measure OVOCs.

**Response:** Sorry for the misunderstanding. It is not rigorous to declare certain compounds to be the "most abundant" VOCs without having measured oxygenated VOCs.

Oxygenated volatile organic compounds (OVOCs) are critical atmospheric ozone and secondary organic aerosol (SOA) precursors and radical sources. Those compounds can contribute significantly to the total VOC burden and the total OH reactivity. As in our previous study, OVOCs accounted for 9-17% of TVOC concentration (Yu et al., 2021; Huang et al., 2022). Other research results show that OVOCs are the dominant groups in urban Beijing, Guangzhou, and Pingyuan city (Li et al., 2015a; Han et al., 2019; Luo et al., 2020). However, OVOCs were not simultaneously measured in this study due to the limitations on available instrumentation. Thus, the description of VOCs concentration characteristics in Section 3.1 stated that they apply only to studies which failed to measure OVOCs. Through literature review, we found that only a limited number of photodegradable OVOC species, such as formaldehyde, acetaldehyde, and acetone, have been measured in the field campaigns in China. (Lu et al., 2013, 2012; Tan et al., 2018, 2019a). Many important photodegradable OVOCs, such as larger aldehydes and ketones, carboxylic acids, nitrophenols, organic peroxides, and multifunctional species, have been rarely quantified accurately in ambient environments. This requires us to strengthen the measurement of unknown active components in future research, or conduct in-depth research with the the model results.

It should be noted that this study calculated only the OH reactivity of the measured species, i.e., the impact of unmeasured species, such as secondary products (oxygenated VOCs and nitrates produced by photochemical reactions) and monoterpenes, were not considered. Previous studies have shown that both undetected primary emissions and unmeasured secondary products could contribute to missing reactivity (Yang et al., 2016). Therefore, in this study we provided a lower limit of speciated OH reactivity. OVOCs play an important role to influence the AOC via OH-initiated degradation. A previous review suggested that missing OH reactivity has often been observed, and the difference compared the measured and calculated was attributed to a lack of measurement data for OVOCs (Yang et al., 2016). Steiner et al. (2008) found that OVOCs accounted for 30 – 50% of the modelled urban VOC reactivity by using the regional Community Multiscale Air Quality model (CMAQ).

In another urban study, OVOCs were found to contribute between 11–24% during summertime in Houston (Mao et al., 2010b). In the CareBeijing-2006 and PRIDE-PRD campaigns, Lou et al., (2010) and lu et al., (2013) reported the significant contributions of OVOCs to OH reactivity based on model simulations. Since OVOCs were not simultaneously measured in four Chinese megacities, the OVOC contributions were simulated with a box model. And OVOCs explained between 20% and 23% to the total OH reactivity (Tan et al., 2019). In summary, OVOCs behaved as a major contributor to the total OH reactivity.

In section 3.6.2, we statemented that the effect of VOCs on O3 formation were calculated from the sum of measured species, and does not involve species that were not measured, such as OVOCs. Therefore, we provided a lower limit of the effect of VOCs on $O_3$ formation in this study.

②Also comparing mixing ratios in ppb introduces a bit of a bias. It may be better to convert to microgram per m3 (the units used to define air quality standards for compounds). The total reactive carbon mass contributed by something heavier(e.g. toluene) is much larger than the mass contributed by ethane even when the toluene mixing ratio is half that of ethane because the molecular mass is so much higher.

**Response:**Thank you for your suggestions. The units have been homogenized into ug/cm$^3$. And relevant figures and tables have also been updated (including Fig.2-3; Table 1; Table S1-S3 and Fig. S2-S3). For the record, some thresholds are obtained by referring to previous references, and their units are fixed. Therefore, when studying the specific ratios (*i/n*-pentane, T/B and VOCs/NOx), the unit used in this paper is still ppbv rather than ug/cm$^3$.

[Figure]

Fig. 2 Monthly changes in the concentrations of the typical VOCs species in Zhengzhou. The upper and lower boundaries of the boxes indicate the 75th and 25th percentiles, respectively; the lines within the boxes mark the median; the whiskers above and below the boxes indicate the 90th and 10th percentiles, respectively.

[Figure]

Fig. 3 Diurnal variations in VOCs compounds measured at Zhengzhou. The upper and lower boundaries of the boxes indicate the 75th and 25th percentiles, respectively; the lines within the boxes mark the median; the whiskers above and below the boxes indicate the 90th and 10th percentiles, respectively.

[Figure]

Fig. S2 Monthly changes in the concentrations of VOCs in Zhengzhou.

[Figure]

Fig. S3 Diurnal variations of VOCs meteorological conditions during the measurements.

③The authors have focused their measurements on compounds that come only from primary emissions and have no secondary photochemical sources. This can be justified for a PMF study, where the focus is source apportionment. The advantage of that approach is that when OVOCs are excluded the PMF will not form several different factors for photochemically formed compounds. Neither will it form factors for air masses with different photochemical age as was seen in some PMF source apportionment studies from China which included OVOCs. The disadvantage of leaving out OVOCS is that it makes no sense at all to talk about ozone formation potential or OH reactivity when the most reactive ozone precursors are missing. These things can only be discussed when the most important contributors, the OVOCs, are included in the study. **Hence with the present data it is better to focus the objective around identifying sources and comparing the pie charts of the total VOCs and of individual compounds with different emission inventories.** If the authors focus on

pollution identification and mitigation and on the question which emission inventories for this region are most accurate then they can write a scientifically sound paper. Their current work is not yet suitable for ACP.

**Response:** Thanks for your suggestions. Comparison with emission inventory studies were supplemented.

This study compared the annual average relative contributions of different sources with results from published emission inventory studies. In the VOCs anthropogenic emission inventories established by Lu et al. (2020), 8 categories were classified in VOCs sources, including stationary combustion, on-road mobile source, non-road mobile source, industrial process, solvent use source, fuel oil storage and transportation, biomass burning and others. Vehicle exhaust made the most abundant source to anthropogenic VOCs in both EI and PMF, accounting for 29.7% and 36.9%, respectively. The differences were observed between the result of EI and PMF, primarily because most of the vehicles considered were in urban areas. The contribution of industrial sources and solvent use source as the second and third largest source in both EI and PMF. However, the contribution of those two sources obtained from PMF results was lower than that the EI. The relative contribution of the combustion sources resolved from PMF was higher than that in the EI, accounting for 9.4% and 4.3%, respectively. Large differences primarily because the uncertainties of activity data obtained from statistical information. The residential energy consumption and emissions were poorly recorded in comparison with other sources (Chen et al., 2016; Tao et al., 2018), leading to higher uncertainties in emission estimations. Thus, it is necessary to estimate emissions of residential fossil fuel combustion through scientific approaches.

[Figure]

Fig.S6 Contribution of each source calculated using PMF and EI.

④My second major concern is with the PMF factor identification which is quite unconvincing for some of the factors and not supported by source samples. Pallavi et al. 2019 (Atmos. Chem. Phys., 19, 15467–15482, 2019 https://doi.org/10.5194/acp-19-15467-2019) showed that factor profiles can be contrasted with samples collected near a source or directly at the tailpipe to validate factor profiles and justify factor identification. For factors for which this should be easy such as vehicular exhaust and coal/biomass burning source profiles should be recorded and plotted together with factor profiles to avoid blaming the wrong sources.

**Response:** Thanks for your suggestions. As shown in Fig. S5, the source profiles derived from the PMF analysis were compared with their sources attributed from the source profiles. The data of the source profiles were derived from a local tunnel experiment and a review of the most recent literature. The source profiles for solvent use correlated most strongly between the two methods (R = 0.84). A study by Jin et al. (2022) identified that low-carbon alkanes (e.g., ethane, propane, and isopentane), alkenes (e.g., ethylene, propene, and 1-butene), and aromatics (e.g., benzene, toluene, and m/p-xylene) were the main groups in the tunnel study, and that vehicle emissions agreed well between this factor and the source profiles (R = 0.59). Different profiles of combustion sources were investigated, and the correlation between the results of NG combustion and the PMF factor was strongest (R = 0.57). The above results are to be expected because Zhengzhou has gradually phased out coal-fired boilers and replaced them with gas boilers in recent years. As for industrial sources, the correlation between

the two methods was 0.43. The C2-C3 hydrocarbons accounted for a high proportion in factor 1, whereas the content of aromatics was lower. Industrial production of nonmetallic mineral products in Zhengzhou is reasonably well developed, which leads to large emissions of ethane and propane. The main contributing source of aromatic hydrocarbons is rubber and plastics. However, the scale of the enterprises involved in their production is far lower than that in the Pearl River Delta; consequently, the emission of aromatic hydrocarbons is lower in Zhengzhou.

[Figure]

Fig.S5 The comparison between source profiles derived from the PMF against their attributed sources from the source profiles (bar is associated with PMF and dot is associated with source profiles).

⑤Among the factors reported, the following factors are likely to be correctly identified: "Biogenic", "LPG/CNG" and "Vehicular exhaust". Based on the source fingerprints published by Hakkim et al. 2021 ATMOSPHERIC ENVIRONMENT: X 11 (2021) 100118 this "vehicular exhaust" fingerprint is plausible for a fleet comprising of a mixture of LPG, CNG and conventional petrol/diesel vehicles. It may be best evaluated against a traffic junction sample since it doesn't carry the signature of any specific tail pipe exhaust.

**Response:** Sorry for the mistake. The vehicle exhaust source was rewritten. Factor 3 was was identified by high percentages of C2-C6 alkanes (i.e., ethane (37.8%), propane

(46.9%), iso-butanes (40.9%), n-butanes (38.5%), n-pentane (26.0%), iso-pentane (31.6%), ), C2-C4 alkanes (i.e.,ethylene (31.4%), propylene (38.4%) and 1-butene (67.4%)) and C6-C8 aromatics (i.e., benzene (21.4%), toluene (30.7%) and m/p-xylene (25.6%)). These components are considered to be typical products of incomplete combustion processes (Baudic et al., 2016; Gaimoz et al., 2011; Liu et al., 2008a; Song et al., 2018). It is reported that i-pentane were usually originated from gasoline evaporation (Mo et al. 2017), and 2-methylpentane and 3-methylpentane are tracers of gasoline vehicle exhausts (Tsai et al., 2003; Liu et al., 2008; Song et al., 2018). A ratio of T/B was 2.0 in this profile, which further confirmed the effect of vehicular emissions (Yao et al. 2021). In addition, the source correlated significantly with CO and $NO_2$ (p<0.01), but not with $SO_2$ (p>0.05). Therefore, source 3 was identified as vehicle exhaust.

⑥The following three factors have may have problems with the factor resolution/factor identification Factor 1 "Industrial source", Factor 2 "solvent use", and Factor 5 "Coal burning + Biomass burning". For Factor 1 "Industrial source" and Factor 5 "Coal burning + Biomass burning" the g-space plot for these two factors needs to be shown. It is possible that the same source is getting split into two separate sources already in the 6-factor solution. In fact, the current paper contains no figure that argues why a 6-factor solution is the most plausible in the present case. Hence it cannot be ruled out that the dataset can only separate 5 factors without the g-space plot for these two factors which share almost identical conditional probability plots and diurnal profiles in Figure 7. If the authors can make a case for these two being two separate factors with the g-space plot, then the two sources are still most of the time colinear, and it is hence likely that both are two industrial sources that are very close to each other.

**Response:** Sorry for the mistake. We updated the factors, and six sources were identified using the PMF model. Choosing the optimal number of factors in the model is important. The number of factors depends on Q (ture)/Q (robust) and Q/Qexpected (Qexp). In theory, Q (ture)/Q (robust) < 1.5 and a value close to 1 is considered reasonable (Ulbrich et al., 2009), and the rate of change of Q/Qexp should be stable

and the ratio should be close to 1 (Baudic et al., 2016; Hui et al., 2019). In this study, the numbers of factors used for the PMF analysis were tested from three to eight, and the optimum six-factor solution with Q/Qexp = 0.94, (Q (ture)/Q (robust) = 1.0) was selected. Additionally, Fpeak values from −1 to 1 with 0.1 intervals were used in the model, and Fpeak = -0.2 was established as the best solution (as shown in Fig. S1)

[Figure]

[Figure]

**Fig. S1** The Q/Qexp and Q (ture)/Q (robust) ratios in different solutions (a); the Q/Qexp ratio for different Fpeak value solutions (b).

⑦Factor 5 cannot be a solid fuel combustion source as solid fuel combustion always produces a set of aromatic hydrocarbons (Benzene, toluene, xylenes) and some other compounds e.g. several alkanes and alkenes at the same time. A source with large quantities of few very specific compounds (ethane, benzene, acetylene in this case) is typically industrial in nature. Benzene and acetylene are the only two compounds that have significantly higher emission factors in any type of flaming combustion with very high combustion efficiency while ethane and the small quantities of other hydrocarbons such as propane in the source profile could be unburned fuel leaking. So, it appears to me that this source profile is from a very specific source that is burning almost pure ethane gas. If there is petrochemical industry around then this could be a gas flare. It is very unlikely that the source is combusting coal of solid biofuel. Solid fuels like coal and biomass burning usually emit a larger set of aromatic hydrocarbons including C8 and C9 aromatics, and larger quantities of alkenes such as ethene and propene. If the authors want to make a case that this is coal burning or solid biofuel burning, they would have to collect source samples to show the source profile. However, many authors have published source profiles for a wide range of combustion in different kind of devices in China. E.g. Yan et al. 2016 Atmospheric Environment Volume 143, 261-269 published source profiles from power plants fired with different types of coal and biomass. Alkenes dominate the emission profiles followed by alkanes and aromatics. Wang et al. 2013 Front. Environ. Sci. Eng. 2013, 7(1): 66–76 showed that alkenes and carbonyls dominated emissions when solid fuels such as coal or biomass briquettes used in residential stoves. Yang et al. 2020 published emission factors for coal and oil boilers Journal of Environmental Sciences Volume 92, June 2020, Pages 245-255. Overall, this factor is incompatible with solid or even liquid fuel and must be caused by some kind of gas burner unless it is the same source as the "industrial source" getting split into two different sources to account for different combustion efficiencies in different parts of the industrial process cycle.

**Response:** Sorry for the mistake. This source were was considered to be fuel burning. Different source profiles of combustion source were investigated, and the correlation between the results of natural gas combustion and PMF factor is more correlated

(R=0.57). The results are expected, because Zhengzhou has gradually banned coal-fired boilers and replaced them with gas in recent years.

- Factor 5 was distinguished by substantial amounts of acetylene (72%), which is a marker of combustion sources (Hui et al., 2021). Additionally, the source was also characterized by considerable amounts of benzene and C2-C3 hydrocarbons, which are representative species of incomplete combustion processes (Zheng et al., 2021). Meanwhile, the independent tracers (i.e., NO2, SO2, and CO) exhibited correlation with this factor (R2 > 0.3, p < 0.01); therefore, source 5 was considered to be fuel burning. Factor 5 displayed high CPF values when the wind was from east. This is possibly related to the heating companies located within 1.0 km to the east of the site. The diurnal variation of this factor was characterized by apparent increase at night, which could be related to the accumulation of pollutants associated with nighttime heating.

⑧The factor identification of Factor 2 Solvent use may be correct but would have to be supported with source profiles. In fact, among all factor profiles in this study, this factor profile is probably the one most compatible with stack sample of coal / biomass fired power plants from Yan et al. 2016 Atmospheric Environment Volume 143, 261-269 or coal/oil fired boilers Yang et al. 2020 Journal of Environmental Sciences Volume 92, June 2020, Pages 245-255. Hence the authors would have to match the source profile against solvent source profiles e.g. from Lui et al. 2008 Atmospheric Environment Volume 42, Issue 25, August 2008, Pages 6247-6260 and the alternative source profiles (coal/biofuel) and would have to decide based on that analysis as well as the conditional probability plots (in which direction from the observational site is Zhengzhou thermal power plant located) in addition to the correlation with other combustion tracers currently used to affirm the identification.

**Response:**Thanks for your suggestions. This factor were rewritten. Factor 2 (F2) was characterized by high n-hexane (60.4%), n-octane (26.3%), methyl cyclohexane (20.4%), o-xylene (79.6%), m/p-xylene (61.7%), p- diethylbenzene (84.0%), o-xylene (79.0%), m-ethyl toluene (57.8%), o-ethyl toluene (29.2%), ethylbenzene (51.1%) and

toluene (20.8%). These compounds are major components emitted from various solvents or industrial processes (Zhou et al. 2019, Wang et al. 2021). However, there are almost no tracers of ethane, ethene, acetylene and benzene related to combustion sources, and this source appeared to exhibit poor correlations with gas tracers ($R^2 < 0.10$). Therefore, F2 was identified as solvent use. And the source profiles for solvent use correlates most strongly between the two methods (R=0.84). The CPF plots of this factor suggested southeast were the dominant source directions. This may be because the large automobile manufacturers in southeast of urban area.

The source profiles derived from the PMF analysis were compared with their sources attributed from the source profiles. The data of the source profiles were derived from a review of the most recent literature. The source profiles for solvent use correlated most strongly between the two methods (R = 0.84).

⑨The language used is often colloquial and inappropriate. The manuscript will have to go through language editing.

**Response:**Thanks for your suggestions. The manuscript had to go through language editing. We thank James Buxton, MSc, from Liwen Bianji (Edanz) (www.liwenbianji.cn/), for editing the English text of a draft of this manuscript.

⑩The information describing the measurements in the methods section is insufficient. What does "continuously monitored" in line 114 mean what is the time resolution of the sampling?

**Response:** Thanks for your suggestions. The description of Line 114-115 has been corrected to "The VOC species are continuously monitored by using an auto-GC system with a 1-h time resolution (AMA Instruments GmbH, Germany).".

11.The authors have conveniently outsourced all their PMF methods section to the supplement. This is not appropriate, certain key points must be described in the main text. Also, the relevant information is incomplete.

**Response:** Thanks for your suggestions. This part has been added to the manuscript in Section 2.2. Certain key points of PMF methods were as follows:

"In this study, analysis of the source of the VOCs was performed using the EPA PMF 5.0 model, which is a receptor model used widely for source apportionment (Gao et al., 2018; Yadav et al., 2019). Detailed information regarding this method is available in the user manual (Norris et al., 2014) and other related literature (Song et al., 2019a, 2019b). Two input files are required for PMF: the concentration values and the uncertainty values of the individual VOC species. The uncertainty is calculated using Eq. (1) when the species concentration value is higher than its method detection limit (MDL), or using Eq. (2) when the concentration is less than or equal to the MDL:

$$\text{Unc} = \sqrt{(\text{EF} \times \text{c})^2 + (0.5 \times \text{MDL})^2}, \quad (1)$$

$$\text{Unc} = \frac{5}{6} \times \text{MDL}, \quad\quad\quad (2)$$

where c is the concentration of the individual VOC species, and EF is the error fraction, which was set to 10% of the VOC concentration (Yuan et al., 2012).

Owing to the complexity of the chemical reactions, not all of the VOC species were used in the PMF analysis. Based on previous work, this study adopted the following principles for selection of the VOC species. (1) Species with more than 25% of data missing or below the MDLs were rejected, which follows the methodology of previous studies (Zhou et al., 2019). (2) Species with short atmospheric lifetimes were excluded because they rapidly react away in the atmosphere. (3) Species that represent source tracers of emission sources were retained (e.g., in the case of isoprene). Eventually, 27 VOC species were selected for source apportionment analysis. VOC species were grouped into strong, weak and bad according to their signal/noise ratio (S/N), and there were 23 and 4 species grouped into strong and weak, respectively. It should be noted that the volumetric concentration (ppbv) of the individual VOC species was converted to mass concentration ($\mu$g m$^{-3}$) before being input into the PMF model.

Choosing the optimal number of factors in the model is important. The number of factors depends on Q (ture)/Q (robust) and Q/Qexpected (Qexp). In theory, Q (ture)/Q (robust) < 1.5 and a value close to 1 is considered reasonable (Ulbrich et al.,

2009), and the rate of change of Q/Qexp should be stable and the ratio should be close to 1 (Baudic et al., 2016; Hui et al., 2019). In this study, the numbers of factors used for the PMF analysis were tested from three to eight, and the optimum six-factor solution with Q/Qexp = 0.94, (Q (ture)/Q (robust) = 1.0) was selected. Additionally, Fpeak values from −1 to 1 with 0.1 intervals were used in the model, and Fpeak = -0.2 was established as the best solution (as shown in Fig. S1)."

12. Information on which species were weak and strong is missing. It appears the authors are running the PMF with mixing ratios in ppb instead of converting everything to microgram per m3 as it should be done. Without converting the input files to mass loadings, it will not be possible to compare the pie charts that can be extracted from the PMF with emission inventories which have the VOC emissions in units of weight.

**Response:**Thank you for your suggestions. In this paper, we are running the PMF with mixing ratios in ug/cm$^3$.   And information on which species were weak and strong were supplemented. VOC species were grouped into strong, weak and bad according to their signal/noise ratio (S/N), and there were 23 and 4 species grouped into strong and weak, respectively.

13. The justification why their VOC dataset can resolve 6 factors in not sufficient. The authors need to show a plot that depicts how the i factors identified by the PMF change from the 3 to the 8-factor solution. Then they need to justify their chosen 6 factor solution (why not 5 factors, why not 7 factors) Can the solution be defended based common sense (are all sources that should be expected there and do their source profiles match with the factor profiles produced by the PMF). Is the g-space plot of the solution OK or problematic? When two factors are linearly corelated in their g-space plot then the PMF cannot resolve that many factors and a solution with less factors must be used.

Overall, the quality of the text is not up to the mark for example the author's state: "In this study, seven-factors are extracted by PMF model based on: (1) principal component analysis of the VOC data; (2) VOC emission inventory of research region based on field

investigation; and (3) Q/Qexp ratio for different factor numbers in the PMF (Fig. S1)."

Yet they actually present a 6-factor solution. I am afraid in the current form the manuscript cannot be considered to be up to the quality benchmark of ACP.

There is no description of the uncertainty analysis performed by the authors. Did they run bootstraps? F-peak analysis? Displacement runs?

**Response:**Thank you for your suggestions. Choosing the optimal number of factors in the model is important. The number of factors depends on Q (ture)/Q (robust) and Q/Qexpected (Qexp). In theory, Q (ture)/Q (robust) < 1.5 and a value close to 1 is considered reasonable (Ulbrich et al., 2009), and the rate of change of Q/Qexp should be stable and the ratio should be close to 1 (Baudic et al., 2016; Hui et al., 2019). In this study, the numbers of factors used for the PMF analysis were tested from three to eight, and the optimum six-factor solution with Q/Qexp = 0.94, (Q (ture)/Q (robust) = 1.0) was selected. Additionally, Fpeak values from −1 to 1 with 0.1 intervals were used in the model, and Fpeak = -0.2 was established as the best solution (as shown in Fig. S1).

[Figure]

[Figure]

**Fig. S1** The Q/Qexp and Q (ture)/Q (robust) ratios in different solutions (a); the Q/Qexp ratio for different Fpeak value solutions (b).

14. I am not quite comfortable with the OH-reactivity analysis given that OVOCs which often contribute close to 80% of the total OH reactivity has not been measured.

**Response:**Thank you for your suggestions. We updated the relevant contents of OH reactivity of measured species. It should be pointed out that the impact of ovoc is deducted in the comparative analysis of other literature (as shown in Table S5).

It should be noted that this study calculated only the OH reactivity of the measured species, i.e., the impact of unmeasured species, such as secondary products (oxygenated VOCs and nitrates produced by photochemical reactions) and monoterpenes, were not considered. Previous studies have shown that both undetected primary emissions and unmeasured secondary products could contribute to missing reactivity (Yang et al., 2016). Therefore, in this study we provided a lower limit of speciated OH reactivity. OVOCs play an important role to influence the AOC via OH-initiated degradation. A previous review suggested that missing OH reactivity has often been observed, and the difference compared the measured and calculated was attributed to a lack of measurement data for OVOCs (Yang et al., 2016). Steiner et al.

(2008) found that OVOCs accounted for 30 – 50% of the modelled urban VOC reactivity by using the regional Community Multiscale Air Quality model (CMAQ). In another urban study, OVOCs were found to contribute between 11–24% during summertime in Houston (Mao et al., 2010b). In the CareBeijing-2006 and PRIDE-PRD campaigns, Lou et al., (2010) and lu et al., (2013) reported the significant contributions of OVOCs to OH reactivity based on model simulations. Since OVOCs were not simultaneously measured in four Chinese megacities, the OVOC contributions were simulated with a box model. And OVOCs explained between 20% and 23% to the total OH reactivity (Tan et al., 2019). In summary, OVOCs behaved as a major contributor to the total OH reactivity.

**Table S5** The OH reactivity towards the total VOCs and the comparison with other studies (unit: $s^{-1}$).

| | The OH reactivity of the total VOCs | The OH reactivity of the total OVOCs | The OH reactivity after deducting OVOCs | References |
|---|---|---|---|---|
| Zhengzhou | 6.7 | - | 6.7 | This study |
| Xianghe | 7.9 | 2.4 | 5.5 | Yang et al., 2020 |
| Beijing | 15.5 | 7.2 | 8.3 | Yang et al., 2021 |
| Heshan | 18.3 | 4.7 | 13.6 | Yang et al., 2017 |
| Shanghai | 6.21 | 2.97 | 3.24 | Tan et al., 2019 |
| Guangzhou | 10.9 | 4.6 | 6.4 | Tan et al., 2019 |
| Chongqing | 8.9 | 2.136 | 6.8 | Tan et al., 2019 |

Minor comments:

Line 29: Can you be more specific on which species control strategies should focus. Don't leave it to policy makers to define "key species" after reading your paper. Tell them which compounds to focus on.

**Response:**Thank you for your suggestions. The concentrations of the key species were markedly different among the four seasons. Ethylene and propylene had highest OH reactivity in winter, which is speculated to be related to the emission of combustion sources. Additionally, isoprene made the largest contribution to the total OH reactivity in both summer and spring, reflecting the substantial effect of biogenic

sources. Therefore, control strategies based on OH reactivity should focus on the key species.

Line 39: You probably mean to say that pollution control policies are working

**Response:**Thanks for your suggestions. We have corrected it. The description of Line 40-41 has been corrected to"The proportion of industrial and solvent sources presented a decreasing trend, which reflects the remarkable effect of control policies."

Line 50: Colloquial language like "is booming" should be avoided in scientific manuscripts

**Response:**Sorry for our carelessness. We have corrected it. The description of Line 52-53 has been corrected to "the study of VOCs is a primary focus among the scientific community and relevant governing bodies".

Line 55: since you only have references from China it may be better to say "in many regions in China". If you are attempting to make a "global" statement then you need to support it with references from different regions around the globe.

**Response:**Thanks for your suggestions. We have supplemented the literature of other countries.

Uttamang, P., Campbell, p., Aneja, V., Hanna, A., 2020.  A multi-scale model analysis of ozone formation in the Bangkok Metropolitan Region, Thailand. Atmos. Environ. 229,117433.

Sadeghi, B., Pouyaei, A., Choi, Y., Rappenglueck, B., 2022. Influence of seasonal variability on source characteristics of VOCs at Houston industrial area. Atmos. Environ. 277, 119077.

Yadav, R., Sahu, L., Tripathi, N., Pal, D., Beig, G., Jaaffrey, S., 2019. Investigation of emission characteristics of NMVOCs over urban site of western India. Environ. Pollut. 252, 245-255.

Line 56: The statement about the ozone formation potential must be qualified. "Studies which do not report OVOCs usually identify aromatics and alkenes as the largest contributors of ozone formation potential (OFP) (Li et al., 2019b; Yan et al., 2017), while other studies typically identify OVOCs as the largest contributor to OFP (Li et al. 2014, Atmospheric Environment Volume 99, December 2014, Pages 403-410)".

**Response:** Thanks for your suggestions. We have corrected it. The description of Line 57-60 has been corrected to "In many regions, alkanes represent the dominant VOC species, while studies which do not report OVOCs usually identify aromatics and alkenes as better contributors of ozone formation potential (OFP) (Li et al., 2019b; Yan et al., 2017)."

Line 110: Typical urban environment is an acceptable statement. No large industrial sources not really. Most of the VOCs measured have lifetimes of > 1 hr. The authors may calculate their fetch region from which air can reach the measurement site within 2-3 hours at the average wind speed. Since that radius would likely be more than 10 km Zhengzhou thermal power plant and several industrial areas would likely fall within the fetch region.

**Response:** Thanks for your suggestions. We have corrected it.

Line 185 ff: This comparison of total VOC mixing ratios is a bit problematic. The comparison should be split into studies that reported only alkanes, alkenes, terpenes and aromatic compounds and studies that reported other additional compounds such as OVOCs and/or halocarbons. The studies that report a large set of VOC classes usually report higher mixing ratios. It is not OK to compare apples with pears without saying which ones are the apples and which ones are the pears. It may be easier to compare mixing ratios of specific VOCs between sites without introducing accidental bias.

Among the observed species, alkanes were the major component of the VOCs with mean concentration of 54.7±37.9 μg/m3, accounting for 58% of the total, followed by aromatics (25%), alkenes (13%) and alkynes (3%). Many previous related studies also reported that alkanes represent the dominant group (Fu et al., 2020; Gu et al., 2020), similar to the situation found in Zhengzhou. For the record, OVOCs were not simultaneously measured in this study due to the limitations on available instrumentation. Thus, those investigations apply only to studies which failed to measure OVOCs.

Line 194 ff: This statement is only valid if the studies cited did measure OVOCs if those studies, just like the present one, did not then it is important to qualify that this applies only to studies which failed to measure OVOCs.

**Response:**Thanks for your suggestions. We have corrected it. The description of Line 219-233 has been corrected to "The average ambient VOC concentrations and chemical species measured in Zhengzhou during the study period are shown in Table S2. And Table S3 presented the comparison of VOCs between this study and other studies. The annual average concentration of VOCs was 94.3±53.1 μg/m3 (38.2± 15.6 ppbv), i.e., close to the concentration reported in Langfang (33.4 ppbv) (Song et al., 2019a) and Nanjing (34.4 ppbv) (Shao, et al. 2016), lower than that found in Chengdu (41.8 ppbv) (Song et al. 2018), Guangzhou (42.7 ppbv) (Zou et al., 2015), and higher than that reported in Tianjin (28.7 ppbv) (Liu et al. 2016). Among the observed species, alkanes were the major component of the VOCs with mean concentration of 54.7±37.9 μg/m3, accounting for 58% of the total, followed by aromatics (25%), alkenes (13%) and alkynes (3%). Many previous related studies also reported that alkanes represent the dominant group (Fu et al., 2020; Gu et al., 2020), similar to the situation found in Zhengzhou. For the record, OVOCs were not simultaneously measured in this study due to the limitations on available instrumentation. Thus, those investigations apply only to studies which failed to measure OVOCs."

**Table S3.** Comparison of VOCs (ppbv) between this study and other studies.

| City | TVOC | alkanes | alkenes | aromatics | alkynes | Reference |
|------|------|---------|---------|-----------|---------|-----------|
| Zhengzhou | 38.2±15.6 | 23.0±19.5 | 7.1±3.3 | 5.5±1.3 | 2.6±2.6 | This study |
| Nanjing | 34.40 ±25.20 | 14.98±12.72 | 7.35 ± 5.93 | 9.06 ± 6.64 | 3.02 ± 2.01 | Shao, et al. 2016 |
| Guangzhou | 44.56 | 26.2 | 7.33 | 11.03 | -- | Zou, et al. 2015 |
| Chengdu | 41.8 ± 20.8 | 23.6 ± 13.0 | 8.2 ± 6.4 | 7.2 ± 6.1 | 2.7 ± 2.3 | Song et al. 2018 |
| Langfang | 33.38 ± 26.03 | 22.93 ± 19.15 | 3.7 ± 2.72 | 4.91 ± 5.7 | 2.56 ± 1.85 | Song, et al. 2019a |
| Tianjin | 28.7 ± 11.4 | 18.3 ± 6.0 | 5.2 ± 2.0 | 5.3 ± 5.9 | -- | Liu et al. 2016 |

**Response:** Thanks for your suggestions. We have corrected it.

Line 198 they do not account for 90% of the VOCs the account for 90% of the compound classes monitored in the present study.

**Response:** Thanks for your suggestions. We have corrected it. The description of Line 234-236 has been corrected to "To clarify the characteristics of VOC emission sources, the concentrations of the 20 most abundant species, accounting for 83% of the compound classes monitored in the present study, are listed in Table 1."

Line 208: Isoprene emissions have also been reported from biomass burning e.g. from smouldering rice straw fires (Kumar et al. 2021 Science of the Total Environment 789 (2021) 148064).

**Response:** Thanks for your suggestions. We have corrected it. The description of Line 247-248 has been corrected to "whereas isoprene is a typical biogenic tracer (Maji et al., 2020). And isoprene emissions have also been reported from biomass burning e.g. from smouldering rice straw fires (Kumar et al., 2021)."

Line 293ff: The authors flag diagnostic emission ratios (e.g. B/T ratios) for different sources and compare them with ambient observations but not with their PMF source fingerprints. Had they compared with their PMF finger prints they would have realized that the "Solvent" factor has a B/T ratio somewhere in between traffic and coal combustion. It makes no sense to compare diagnostic ratios to ambient mixing ratios which represent a source mixture instead of comparing them to individual PMF

factor profiles (which should have emission ratios that match source fingerprints) in a paper presenting a PMF analysis.

Line 304ff: Again diagnostic I-pentane to n-pentane ratios should be compared to source fingerprints not to ambient mixing ratios.

**Response:**Thanks for your suggestions. Ratios analysis by using ambient mixing ratios is a common research method (Hu, et al. 2018; Li, et al. 2020; Song, et al. 2019; Yang, et al. 2018; Zhou, et al. 2019). And it is widely used in air pollution research to make initial judgements on the sources of atmospheric VOCs and the degree of ageing of air masses.

Several studies have shown that the composition characteristics of VOCs released by one source were relatively stable and different from those of other sources (Jobson et al., 1999; Barletta et al., 2005). According to correlations and the characteristic ratios between the representative pollutants, the source of ambient VOCs can be crudely deduced. Thus, his study adopted the benzene/toluene (B/T) ratio and the i-pentane/n-pentane ratio as the preferred metrics.

Hu, R., et al.2018     Levels, characteristics and health risk assessment of VOCs in different functional zones of Hefei. Ecotoxicol Environ Saf 160:301-307.

Li, Jie, et al.2020Characteristics, sources and regional inter-transport of ambient volatile organic compounds in a city located downwind of several large coke production bases in China. Atmospheric Environment 233.

Song, Mengdi, et al.2019 Sources and abatement mechanisms of VOCs in southern China. Atmospheric Environment 201:28-40.

Yang, X., et al.2018  Characterization of volatile organic compounds and the impacts on the regional ozone at an international airport. Environ Pollut 238:491-499.

Zhou, X., et al.2019   Volatile organic compounds in a typical petrochemical industrialized valley city of northwest China based on high-resolution PTR-MS measurements: Characterization, sources and chemical effects. Sci Total Environ 671:883-896.

Section 3.3.4: CPF have no meaning if not related to sources around the site. Which side of the monitoring site is the local coal fire power plant located? Refinery? Industrial areas? Forests? This type of analysis makes sense with a local map (e.g. Google Earth) in which relevant features are labelled. Not without.

**Response:**Thanks for your suggestions. The description of CPF has been updated.

Section 3.5.1. Since OVOCs which contribute most to the total OH reactivity in most sites around the world were not monitored, I would recommend removing this analysis. If retained it must be qualified that several very reactive compounds that contribute more than half of the total OH reactivity at some sites in China where they were measured (Aldehydes) were not included in this current analysis.   Again while comparing with the OH reactivity reported in other studies one has to be careful and must group into studies that reported only the same functional groups that were reported in the present study and studies that include OVOC and/or halogenated compounds.

   **Response:**Thanks for your suggestions. Table S4 shown the OH reactivity towards the total VOCs and the comparison with other studies. It should be noted that we only compare the part deducting the impact of OVOCs. It should be noted that this study calculated only the OH reactivity of the measured species, i.e., the impact of unmeasured species, such as secondary products (oxygenated VOCs and nitrates produced by photochemical reactions) and monoterpenes, were not considered. Previous studies have shown that both undetected primary emissions and unmeasured secondary products could contribute to missing reactivity (Yang et al., 2016). Therefore, in this study we provided a lower limit of speciated OH reactivity.

**Table S4** The OH reactivity towards the total VOCs and the comparison with other studies (unit: s$^{-1}$).

| | The OH reactivity of the total VOCs | The OH reactivity of the total OVOCs | The OH reactivity after deducting OVOCs | References |
|---|---|---|---|---|
| Zhengzhou | 6.7 | - | 6.7 | This study |
| Xianghe | 7.9 | 2.4 | 5.5 | Yang et al., 2020 |
| Beijing | 15.5 | 7.2 | 8.3 | Yang et al., 2021 |
| Heshan | 18.3 | 4.7 | 13.6 | Yang et al., 2017 |
| Shanghai | 6.21 | 2.97 | 3.24 | Tan et al., 2019 |
| Guangzhou | 10.9 | 4.6 | 6.4 | Tan et al., 2019 |
| Chongqing | 8.9 | 2.136 | 6.8 | Tan et al., 2019 |

Section 3.5.2. has the same problem as section 3.5.1. Since OVOCs were not monitored the relative rating of the contribution to the OFP is very biased. I would recommend removing this analysis.

**Response:**Thanks for your suggestions. We statemented that the effect of VOCs on O3 formation were calculated from the sum of measured species, and does not involve species that were not measured, such as OVOCs. Therefore, we provided a lower limit of the effect of VOCs on O$_3$ formation in this study.

---

## Author Response (AR2)

**Itemized Response to Reviewer's Comments**

**Ms. Ref. No.:** ACP-2021-1016

**Title:** Measurement report: Intra/interannual variability and source apportionment of VOCs during 2018–2020 in Zhengzhou, Central China

1.Some factors have been renamed in the text but not in the corresponding figures. In some figures the factor names are updated and in others not.

Figure 5 has both old and new factor names in the same figure. In addition it has two version of supposedly the same? conditional probability analysis (one with a line underneath the corresponding BW plot and one in filled at the bottom). Unfortunately the overall shape of the curve appears to be very different between the two versions for the same factor. E.g. on the top the biogenic factor has a 0.35 conditional probability to the north (0 to 30 degree) and also towards 150 degree. Yet in the bottom version the conditional probability towards both these directions is < 0.25. Similar problems exist in all panels. Which version is the reader supposed to believe? Are these for different percentiles? For different model runs?

**Response:** Sorry for the mistake. Figure 1 was drawn using the updated PMF results. Combined with the related sources around the site, the latest version is credible.

The inconsistency in Fig.5 is given by a different PMF run. To better identify each pollution source, the factors are constrained by using tool of toggle constraint in the PMF model based on your constructive comments. After adjustment, the dQ value is still within a reasonable range, which proves that the output result is reasonable. The constrained results with new factor profile are more relevant to the local source profiles and emission inventory, so they can better reflect the actual situation of the local atmosphere.

Meanwhile, we have done the following work, which may affect the results of CPF and PMF.

1. Reject 0 value: Due to instrument failure, some values of wind speed and direction were recorded as 0, which were not excluded in previous studies. In addition, some values in the time series of the PMF results are negative. The above abnormal values were eliminated in the latest study.

2. Unit of PMF: The reviewer mentioned that the unit of PMF should be microgram per $m^3$ instead of ppbv. Thus, we are running the PMF with mixing ratios in $\mu g/cm^3$.

3. PMF model settings: The technical guide for source analysis of ozone pollution in ambient air requires the error fraction (EF) of VOCs to be

within 30%. The EF value used in previous studies was 30%, while it was set to 10% of the VOC concentration in the latest studies. Meanwhile, information on which species were weak and strong is missing. We have corrected it. VOC species were grouped into strong, weak, and bad according to their signal/noise ratio (S/N), and there were 22 and 4 species grouped into strong and weak, respectively.

[Figure]

Fig. 1 Directional dependence and hourly record of each source in Zhengzhou.

2.The Q ratios which the authors use to argue the validity of their solution are not a reliable quality indicator when used in isolation. Specifically, they cannot identify a solution with too much rotational ambiguity. In general, the better the Q ratio the higher the risk of having a solution with too much rotational ambiguity. Such solutions also often cause problems in the bootstrap runs (unmapped factors or poor mapping of bootstrap factors on the original factor). Rotational ambiguity can only be assessed with the help of G-space plots or a cross correlation analysis of the factor contribution time series of all factors. Either the G-space plot or the R of the factor pairs needs to be provided. If any factor pair has an R >0.6 then this is a strong indication that there is too much rotational ambiguity and that the same source is getting split into two factors.

**Response:** Thank you for your suggestions. Choosing the optimal number of factors (P-value) is a critical question in PMF analysis. According to previous studies (Baudic, et al. 2016; Hui, et al. 2020; Liu, et al. 2020; Song, et al. 2019; Wang, et al. 2021; Zheng, et al. 2018), the ratios of Q (ture)/Q (robust) and Q/Qexpected (Qexp) were tested to determine optimum solution.

The G-Space Plot screen (Figure 2) shows scatter plots of one factor

versus another factor, which can be used to assess the relationship between source contributions. The results show that there is no correlation between different factors. The G-space scatter diagram is evenly distributed and the edge is parallel to the X and Y axes, and the result is reliable.

[Figure]

[Figure]

Fig. 2 The G-Space Plot by using PMF model.

Baudic, Alexia, et al.2016 Seasonal variability and source apportionment of volatile organic compounds (VOCs) in the Paris megacity (France). Atmospheric Chemistry and Physics 16(18):11961-11989.

Hui, Lirong, et al.2020 VOC characteristics, chemical reactivity and sources in urban Wuhan, central China. Atmospheric Environment 224.

Liu, Y., et al.2020 Characterization and sources of volatile organic compounds (VOCs) and their related changes during ozone pollution days in 2016 in Beijing, China. Environ Pollut 257:113599.

Song, Mengdi, et al.2019 Sources and abatement mechanisms of VOCs

in southern China. Atmospheric Environment 201:28-40.

Wang, M., et al.2021  Impact of COVID-19 lockdown on ambient levels and sources of volatile organic compounds (VOCs) in Nanjing, China. Sci Total Environ 757:143823.

Zheng, Huang, et al.2018 Monitoring of volatile organic compounds (VOCs) from an oil and gas station in northwest China for 1 year. Atmospheric Chemistry and Physics 18(7):4567-4595

3.Furthermore, the authors should upload the dataset used for their study in a public repository (e.g. Zenodo), and cite the corresponding doi.

**Response:** The dataset has been uploaded to the public repository (Zenodo), and the link is

https://zenodo.org/record/6815259#.Ysq_lnZBy3A .